# GAIA: Rethinking Action Quality Assessment for AI-Generated Videos

**Zijian Chen**[1], **Wei Sun**[1*], **Yuan Tian**[1], **Jun Jia**[1], **Zicheng Zhang**[1],
**Jiarui Wang**[1], **Ru Huang**[2], **Xiongkuo Min**[1*], **Guangtao Zhai**[1*], **Wenjun Zhang**[1]
[1]Shanghai Jiao Tong University  [2]East China University of Science and Technology
[*]Corresponding authors
https://github.com/zijianchen98/GAIA

## Abstract

Assessing action quality is both imperative and challenging due to its significant impact on the quality of AI-generated videos, further complicated by the inherently ambiguous nature of actions within AI-generated video (AIGV). Current action quality assessment (AQA) algorithms predominantly focus on actions from real specific scenarios and are pre-trained with normative action features, thus rendering them inapplicable in AIGVs. To address these problems, we construct **GAIA**, a **G**eneric **AI**-generated **A**ction dataset, by conducting a large-scale subjective evaluation from a novel causal reasoning-based perspective, resulting in 971,244 ratings among 9,180 video-action pairs. Based on GAIA, we evaluate a suite of popular text-to-video (T2V) models on their ability to generate visually rational actions, revealing their pros and cons on different categories of actions. We also extend GAIA as a testbed to benchmark the AQA capacity of existing automatic evaluation methods. Results show that traditional AQA methods, action-related metrics in recent T2V benchmarks, and mainstream video quality methods perform poorly with an average SRCC of 0.454, 0.191, and 0.519, respectively, indicating a sizable gap between current models and human action perception patterns in AIGVs. Our findings underscore the significance of action quality as a unique perspective for studying AIGVs and can catalyze progress towards methods with enhanced capacities for AQA in AIGVs.

## 1 Introduction

Action quality assessment (AQA), which aims to quantify *how well* actions are performed, is a growing area of research across various domains (*e.g.*, [77, 58, 76, 27, 96, 100]). It is becoming especially challenging since generative models like Sora [68, 73] have revolutionized the creation of visually realistic videos. Assessing how well an action is presented can be difficult because of the inherent difference between real videos and generated videos [21, 67]. At minimum, a well-performed action should correctly contain all relevant objects as well as the action subject with recognizable motion presentation while conforming to the physical world dynamics [39, 117]. Moreover, the exponential growth of text-to-video (T2V) models has also given rise to formidable challenges in the evaluation of video action quality, underscoring the increasing need for reliable solutions.

However, there is a significant gap in the existing AQA research. First, prior work has contributed multiple AQA datasets, which predominantly focus on *domain-specific* actions from real videos and collect coarse-grained *expert-only* human ratings [77, 119, 76] on limited dimensions. Meanwhile, the content discrepancies in those AQA videos are often subtle, as the action subjects typically perform similar actions within a consistent environment. Examples include swimming and diving in a natatorium or gymnastics in a gym, which lacks consideration for scene diversity. Second, the

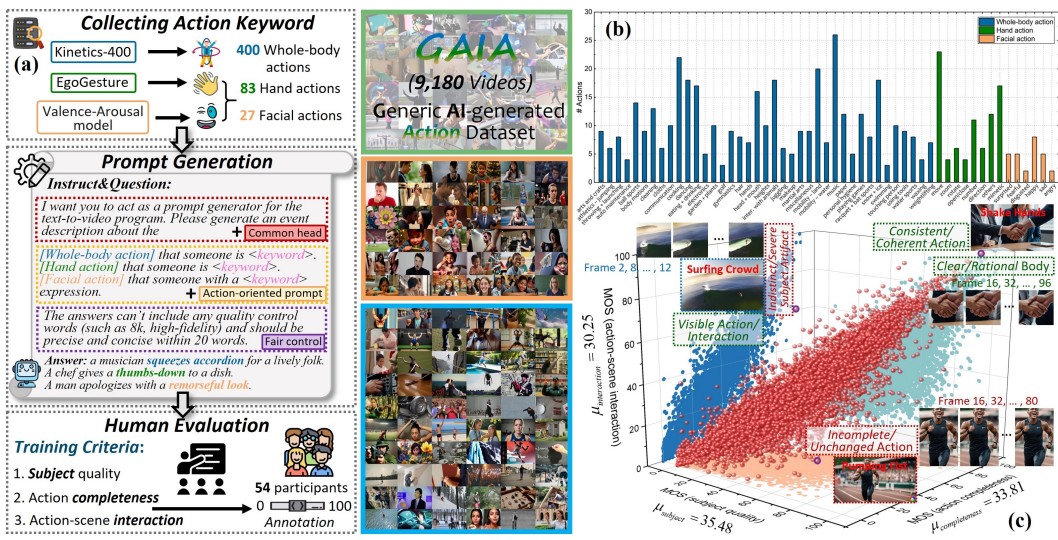

Figure 1: **Data construction pipeline and content overview of GAIA.** (a) Curation process of the GAIA dataset, resulting in 9,180 videos with 971,244 human ratings. (b) The distribution of unique actions per class. (c) 3D scatter plot of the mean opinion score (MOS) in three dimensions and video examples with diverged scores.

existing AQA approaches mainly follow a pose-based or vision-based feature extraction, aggregation, and score regression ternary form, which usually adopt powerful 3D backbone networks that are pre-trained on large action recognition datasets [13, 94] for better feature migration. Nevertheless, a distinguishing characteristic of generated videos is that they may contain atypical actions with various body or object artifacts over time [23, 69], such as aberrant limb count, irrational object shape, and physically implausible motion, due to the stochasticity and unstable nature of the diffusion process. In such cases, the model learned from real action videos may fail in AIGVs with worse prediction performance. At present, it remains unclear to what degree any T2V model can achieve visually rational action generation that varies in action categories, much less the cognitive mechanism of action quality that affects human perception.

To address these issues, we present **GAIA**, a **G**eneric **AI**-generated **A**ction dataset encompassing 9,180 AI-generated videos from 18 T2V models, spanning both lab studies and commercial platforms, which covers a variety of whole-body, hand, and facial actions. Specifically, we recruit 54 participants and conduct a large-scale human evaluation to evaluate the action quality *first-of-this-kind* from three causal reasoning-based perspectives: subject quality, action completeness, and action-scene interaction. Among them, as the major premise of an action, the quality of the action subject directly affects the whole action process. Assessing action completeness ensures that the generated action is not only temporally coherent but also logically and narratively complete. Action-scene interaction considers the spatial relationships, environmental factors, and interactions with other elements within the scene that can influence the perception of the action's quality and realism. Crucially, it provides quantifiable action state estimations based on the behavior of human reasoning in perceiving an action. In theory, this makes complicated coupled action quality approachable and tractable. In practice, the full potential of multi-dimension methods remains largely untapped due to a scarcity of existing datasets, exacerbated by the difficulty of obtaining reliable group subjective opinions. This complements earlier research, which primarily concentrated on AQA under a single real scenario and lacked rating granularity.

We prove the value and type of insights GAIA enables by using it to evaluate the action generation ability and weaknesses across different categories of 18 representative T2V models (several times more than existing benchmarks [23, 51, 69, 67]). Moreover, we contribute a holistic benchmark based on GAIA, which reveals that the existing AQA methods and action-related automatic metrics even video quality assessment (VQA) approaches, correlate poorly with human evaluation. Overall, our study could serve as a pilot for future endeavors aimed at developing accurate AQA methods in generative scenarios while providing substantial insights for better defining the quality of AIGVs.

Table 1: Comparison of **GAIA** and existing AQA datasets. **SS** indicates the source of scores. *Mix* indicates that the participants in human evaluation are recruited across different backgrounds.

| Dataset | Source | Action | Samples | Duration | Avg.Dur. | Resolution | FPS | SS |
|---|---|---|---|---|---|---|---|---|
| MIT Dive (2014) [80] | $Real_{diving}$ | – | 159 | 0.25h | 6.0s | $320\times240$ | 30 | Judge |
| UNLV Dive (2017) [79] | $Real_{diving}$ | – | 370 | 0.4h | 3.8s | $320\times240$ | 30 | Judge |
| AQA-7-Dive (2019) [77] | $Real_{diving}$ | – | 549 | 0.6h | 4.1s | $320\times240$ | 30 | Judge |
| MTL-AQA (2019) [78] | $Real_{diving}$ | – | 1,412 | 1.5h | 4.1s | $1920\times1080$ | 25 | Judge |
| Rhyth. Gym. (2020) [119] | $Real_{gymnastics}$ | – | 1,000 | 26.3h | 95s | $1280\times720$ | 25 | Judge |
| FSD-10 (2020) [66] | $Real_{skating}$ | 10 | 1,484 | – | 3-30s | $1080\times720$ | 30 | Judge |
| Fitness-AQA (2022) [76] | $Real_{workout}$ | 3 | 13,049 | 14.9h | 4.1s | $480^2$-$720^2$ | 30 | Expert |
| FineDiving (2022) [115] | $Real_{diving}$ | 52 | 3,000 | 3.5h | 4.2s | $256\times256$ | 15 | Judge |
| LOGO (2023) [122] | $Real_{swimming}$ | 12 | 200 | 11.3h | 204s | $1280\times720$ | 25 | Judge |
| **GAIA** | **AI-generated** | **510** | **9,180** | **7.1h** | **2.8s** | $\mathbf{256^2}$**-**$\mathbf{2048^2}$ | **4-50** | *Mixture* |

## 2 Related Work

**Action Quality Assessment.** Action quality assessment, which aims to discriminate and evaluate how well an action is performed, has been widely explored in applications such as sports events [95, 80, 79, 77, 78, 72, 58], healthcare [121, 35, 108, 76, 27], and public security [40, 96, 100]. Early works [80, 105] solely consider human pose-based features while neglecting the relations among joints and other action quality-related visual cues (*e.g. splash* in diving or *barbell position* in weightlifting). Later, researchers introduced graph structure to model the joint motion information spatially and temporally [75, 11, 30, 122]. In another line, vision-based approaches [79, 114, 115, 122, 124, 27] combined 3D convolutional neural networks (C3D) [101] with context-aware modules to extract motion-oriented spatial-temporal visual features for assessing action quality. As shown in Tab. 1, there are numerous datasets for AQA, which predominantly focus on the sports domain from real scenarios, while the problem of considering action quality in AI-generated videos remains unexplored. Furthermore, human activities often occur in specific scene contexts, *e.g.*, swimming in a swimming pool. However, it is also possible to enjoy champagne in the pool. Training an action quality assessment model for more diverse AI-generated actions using existing video datasets thus inevitably introduces such bias, which may opposed to paying attention to the actual action in the scene. Choi *et al.* [24] proposed to mitigate scene bias by adding an adversarial loss for scene types and masking out the human actors. Similar operations include extracting the foreground and background parts of the video as data augmented pairs to improve the accuracy of action recognition [38]. Apart from pixel space augmentation, Gorpincenko *et al.* [37] extended it further to utilize the time domain to perform deeper levels of temporal perturbations, thus improving the robustness of action classifiers. The above studies illustrate the necessity of disentangling action process from scenes or decomposing the action process to mitigate the effect of representation bias.

From the perspective of data provenance, virtual worlds and game engines are plausible techniques to generate editable actions as synthetic data before the Generative AI Era. Such synthetic data has been used to train visual models for lots of computer vision tasks (*e.g.*, object detection, recognition, pose estimation, and scene understanding) to extract visual priors. Desouza *et al.* [82] proposed a diverse, realistic, and physically plausible dataset of human action videos using virtual world simulation software. Experiments show that mixing both synthetic and real samples at the mini-batch level during training can significantly improve action recognition accuracy. Similarly, the controllability of both the type and quality of AI-generated actions (using different types of prompts and video generation models) offers a feasible solution for constructing large-scale action quality assessment datasets, especially for those irrational scenarios.

**Video Generative Models.** The recent breakthroughs of generative models [28, 83] expedites massive works towards video generation [49, 53, 110, 15, 120, 71, 41, 109, 43]. As a pioneer, VDM [48] extended the standard image diffusion architecture and presented a 3D U-Net structure [84] to jointly learn the spatial and temporal generation knowledge. Its subsequent works such as Imagen video [46], LaVie [110], and Show-1 [120] cascaded VDM to perform text-conditional video generation and spatial-temporal super-resolution in sequence. AnimateDiff [41] built a flexible MotionLoRA to learn transferable motion priors that can be integrated into text-to-image (T2I) models for video generation. Parallelly, VideoPoet [55] incorporated a mixture of multimodal generative objectives via an autoregressive transformer so as to handle various video generation tasks. More recently, a multi-agent based video generation framework, Mora [118] has been proposed that combines several

Table 2: Summary of popular video generation models: from *open-source* lab studies to large-scale *commercial* creation platforms. We tested the average generation speed (seconds/item) on an NVIDIA RTX4090 locally, except for those closed-source models. OOM is the abbreviation of *out-of-memory*. [†]We report the online generation speed under free plan.

| Model | Year | Mode | Resolution | FPS | Length | Speed | Feature | Open Source |
|---|---|---|---|---|---|---|---|---|
| CogVideo [49] | 22.05 | T2V | 480×480 | 8 | 4s | 12s | − | ✓ |
| Text2Video-Zero [53] | 23.03 | T2V | 512×512 | 4 | 2s | 21s | Pose/Edge Ctrl | ✓ |
| ModelScope [106] | 23.03 | T2V | 256×256 | 8 | 2s | 6s | − | ✓ |
| ZeroScope$_{v2\text{-}576w}$ [8] | 23.06 | T2V | 576×320 | 8 | 3s | 20s | − | ✓ |
| LaVie [110] | 23.09 | T2V | 512×320 | 8 | 2s | 14s | Interpol./Super Res. | ✓ |
| VideoCrafter1 [15] | 23.10 | T2V, I2V | 512×320 | 8 | 2s | 41s | − | ✓ |
| | 23.10 | T2V, I2V | 1024×576 | 8 | 2s | *OOM* | − | ✓ |
| Show-1 [120] | 23.10 | T2V | 576×320 | 8 | 4s | 231s | − | ✓ |
| Hotshot-XL [71] | 23.10 | T2V | 672×384 | 8 | 1s | 14s | Personalized | ✓ |
| AnimateDiff [41] | 23.12 | T2V, I2V | 384×256 | 8 | 2s | 10s | Cam. Ctrl | ✓ |
| VideoCrafter2 [16] | 24.01 | T2V, I2V | 512×320 | 8 | 2s | 45s | − | ✓ |
| Mora [118] | 24.03 | T2V, I2V, V2V | 1024×576 | 25 | >12s | *OOM* | Multi-Agent | ✓ |
| Gen-1 [32] | 23.02 | V2V | 768×448 | 24 | 4s | 52s[†] | Style | − |
| Genmo [2] | 23.10 | T2V, I2V | 2048×1536 | 15 | 4s | 60s[†] | Style, Cam. Ctrl | − |
| Gen-2 [1] | 23.12 | T2V, I2V | 1408×768 | 24 | 4s | 140s[†] | Mot./Cam. Ctrl | − |
| Pika [6] | 23.12 | T2V, I2V, V2V | 1088×640 | 24 | 3s | 45s[†] | Mot./Cam. Ctrl, Sound | − |
| NeverEnds [5] | 23.12 | T2V, I2V | 1024×576 | 10 | 3s | 260s[†] | − | − |
| MoonValley [3] | 24.01 | T2V, I2V | 1184×672 | 50 | 4s | 386s[†] | Style, Cam. Ctrl | − |
| Morph Studio [4] | 24.01 | T2V, I2V | 1920×1080 | 24 | 3s | 196s[†] | Mot./Cam./fps Ctrl | − |
| Stable Video [7] | 24.03 | T2V, I2V | 1024×576 | 24 | 4s | 125s[†] | Style, Mot./Cam. Ctrl | ✓ |

advanced visual AI agents to achieve high-quality, long-form video generation. In addition to the above laboratory studies, several derived commercial video generation products, *e.g.*, Gen-2 [1], Genmo [2], Pika [6], Neverends [5], MoonValley [3], Morph [4], Stable Video [7], and Sora [73, 68], have harvested widespread attention from both academia and industry, exhibiting great possibilities for future AI-assisted video creation.

**Evaluations on Video Generative Models.** Early video generation models shared the same frame-wise evaluation metrics as T2I models, such as Inception Score (IS) [87], Fréchet Inception Distance (FID) [45], and CLIPScore [81], as well as their variants for video [103, 104, 86]. These metrics are all group-targeted and not suitable for assessing a single video. For text-to-video (T2V) models, several benchmarks [23, 67, 51, 69, 57] have been proposed to assess various perspectives like video fidelity [57], temporal quality [67], text-video alignment [51, 69]. Despite covering various dimensions, these works lack specificity and breadth with limited model exploration and human group annotation. Our work differs from current research in three key aspects: 1) We created 510 distinct action prompts covering both coarse-grained and fine-grained actions, each applied with 18 T2V models for extensive assessment. 2) Our casual reasoning-based and multi-dimensional action quality evaluation offers valuable and comprehensive insights into video generation. 3) We have quantitatively validated a large amount of existing metrics that none of them performs well on the AI-generated action quality assessment task.

## 3 Dataset Acquisition

### 3.1 Data Collection

**Prompt Sources.** The marvelous interrelation and working mechanism of body, hand, and face have a high degree of inner unity, which together constitute the key elements of actions [31]. Hence, we sampled action keywords for GAIA from a variety of sources, including the Kinetics-400 [14] for whole-body actions, the EgoGesture dataset [123] and the *valence-arousal* model of affect [85] for fine-grained local hand and facial actions, respectively (Fig. 1(b)). Besides, to avoid linguistic bias and ensure each action keyword appears explicitly in the prompt, we leverage the GPT-4 [9] to design an assembled prompt strategy (Fig. 1(a)). It consists of a *common head*, an *action-oriented description*, and an *output control*, where we intentionally leave out specialized suffixes such as *8k*, *HDR*, *photographic*, and *high fidelity* for fairness. In the meantime, an expert review of the generated prompts is organized to examine the hallucination problem of large language model (LLM) while avoiding NSFW issues for ethical concerns. At last, we obtain 510 prompts for all action categories with an average length of 8.25 words.

**Text-to-Video Models.** To evaluate the action quality of AI-generated videos thoroughly, we select **18** representative T2V models for generation including: 1) **11** open-sourced lab studies: Text2Video-Zero [53], ModelScope [106], ZeroScope [8], LaVie [110], Show-1 [120], Hotshot-XL [71], AnimateDiff [41], VideoCrafter1 (resolution at 512×320 and 1024×576) [15], VideoCrafter2 [16], and Mora [118]; 2) **7** popular commercial creation applications: Gen-2 [1], Genmo [2], Pika [6], NeverEnds [5], MoonValley [3], Morph Studio [4], and Stable Video [7], shown in Tab. 2. Note that CogVideo [49] and Gen-1 [32] are excluded due to the language and mode restrictions. Since we focus on human-centric actions in this paper, other settings such as camera motions or styles are set by template. At last, **9,180** videos were collected. *We defer more details to the Appendix (Sec. B.1).*

## 3.2 Task Definition: the Action Syllogism

Considering the peculiar characteristics of AI-generated videos, to collect a more explainable and nuanced understanding of public perception on action assessment, instead of collecting professional skill scores as in existing AQA studies [76, 115], we opt to collect annotations from a novel perspective, namely the causal reasoning *syllogism* [93, 54]. Specifically, we decompose an action process into three parts: 1) action subject as *major premise*, 2) action completeness as *minor premise*, and 3) interaction between action and scenes as *conclusion*, according to the syllogism theory. The rationale for this strategy is as follows: **(a)** The visibility of the action in videos is greatly affected by the rendering quality of the action subject, which is a crucial element of visual saliency information, while humans excel at perceiving such generated artifacts [21, 51, 70]. **(b)** Moreover, unlike *parallel-form* feedbacks, the order of these three parts in action syllogism inherently aligns with the human reasoning process. For instance, while human annotators are shown with an action scene about "*A musician is playing the piano*", they can intuitively reason like: ❶ a musician as the major premise, which is the subject to execute the action of *playing the piano*; ❷ appearance or completeness of action as the minor premise, containing the spatial and temporal boundaries in the given scenario; ❸ phenomenon of the keys being pressed as the conclusion describing a reasonable result considering both constraints. This reasoning-form evaluation has many merits. First, by breaking down an action into its constituent parts, researchers can more clearly identify and analyze the specific elements that contribute to the perceived quality of the action. Second, such causal reasoning-based strategy is inherently aligned with human perception and can help in understanding how different parts of action are perceived by the public, which can lead to insights into what makes AI-generated action convincing or unconvincing. Third, this scheme allows for a comparative analysis of AI-generated action against natural human action, revealing where AI excels and where it may need improvement.

## 3.3 Subjective Action Quality Assessment

**Participants and Apparatus.** To ensure the comprehensiveness, fairness, and reliability of the evaluation, we recruit a total of **54** participants to participate in our human evaluation, as shown in Tab. 3. All with normal (corrected) eyesight. Considering the viewing effect, a 27-inch Lenovo monitor with a resolution of 2560×1440 is used for video display. The viewing distance and optimal horizontal viewing angle are set as 1.9 times the height of the display ($\approx$ 70cm) and [31°, 58°], respectively. Before the annotation, we instructed all participants to have a clear and consistent understanding of all evaluated aspects and tested their eligibility via a 30-video pre-labeling [20]. In the tutorial for each dimension, participants are guided to rate 10 generated-real video pairs with the same caption. Their answer is compared with ground-truth ratings that were developed by multiple experts. Raters needed to achieve at least 75% ratings that satisfied $|ground\_truth - rating| < 1.5\sigma_{expert}$ to move on to the formal study.

**Main Process.** We adopted a single-stimulus methodology in this evaluation and asked participants to focus on the given action keyword as well as the corresponding prompt and evaluate three action-related dimensions of AI-generated videos, *i.e.*, *subject quality*, *action completeness*, and *action-scene interaction*, by dragging the slide button at a $[0, 100]$ continuous rating scale. We randomly divided the 9,180 videos in GAIA into 31 sessions, with each session, except the last, comprising 300 videos. Ten

Table 3: **Statistics of participants.** *w/* AIGC and *w/o* AIGC denote participants who have or do not have used AI generation tools, respectively.

| Category | Gender | | Background | | Age |
| --- | --- | --- | --- | --- | --- |
| | Male | Female | *w/* AIGC | *w/o* AIGC | |
| **Number** | 39 | 15 | 25 | 29 | 23.4±2.6 |

*golden videos* with expert opinions from the real-world action database [14] were added to each session as an inspection to control the scoring deviations. Only participants who had a high agreement

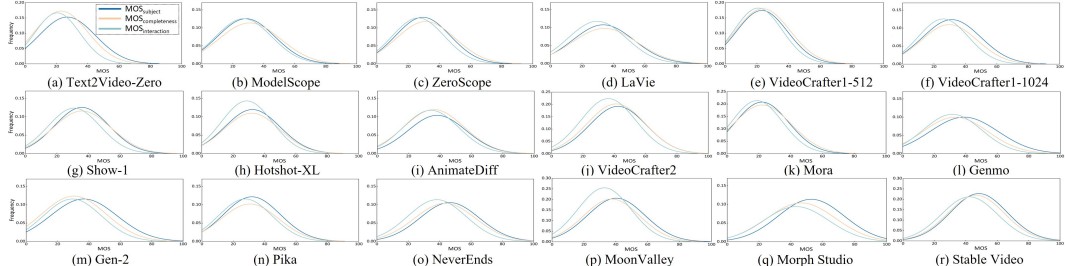

Figure 3: **MOS distributions across different models in terms of subject quality, action completeness, and action-scene interaction.** 11 Lab studies: (a)-(k); 7 Commercial applications: (l)-(r).

(Pearson linear correlation coefficient, $PLCC > 0.7$) with the mean opinion score (MOS) from experts were eligible to continue to the next session, leaving 48 remaining. To reduce visual fatigue, there is a rest segment with at least 15 minutes per 150 videos [90, 21, 19]. In summary, it took participants approximately 2.6 hours to finish one session, and all experiments took over a month to complete. Each participant was compensated $12 for each session according to the current ethical standard [92].

**Quality Control.** In addition to the above pre-labeling and in-process check trial, we noticed 5 line clickers (all male) with over 40% of the same ratings. We removed all their ratings from GAIA dataset. Besides, we follow Otani *et al.*'s [74] recommendation that uses the inter-annotator agreement (IAA) metric (Krippendorff's $\alpha$ [42]) to assess the quality of ratings, where Krippendorff's $\alpha$ for *subject quality*, *action completeness*, and *action-scene interaction* perspectives are 0.6771, 0.6243, and 0.6311, respectively, indicating appropriate variations among annotators. We further calculated the SRCC score using bootstrapping as in KonIQ-10k [50]. Fig. 2 shows the mean agreement (Spearman rank-order correlation coefficient, SRCC) between the MOS values as the number of observers grows. When considering the correlation between nearly 70% of the participants in our study, the mean SRCC reaches remarkably high values of 0.9556,

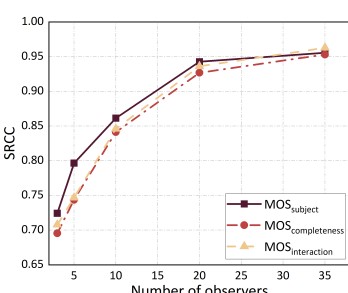

Figure 2: SRCC between MOSs as the observers increases.

0.9531, and 0.9627 in terms of *subject quality*, *action completeness*, and *action-scene interaction*, respectively, which provides a reasonable reference population size for subsequent subjective AQA studies. At last, we obtained a total of **971,244** reliable ratings with an average of **105.8** ratings per video (**35.27** per dimension). We then perform Z-score normalization to the raw MOS of each subject to avoid inter-annotator scoring biases. Here, we abbreviate the MOS of three perspectives as $\mathrm{MOS}_s$, $\mathrm{MOS}_c$, and $\mathrm{MOS}_i$ according to their initials for simplicity. A higher value indicates superior performance or quality in that particular aspect.

### 3.4 Dataset Statistics and Analysis

**Overall Observations.** Each data sample in GAIA consists of four elements: the action keyword $k$, the corresponding prompt $t$, the generated video $v$, and the action quality-related human annotations $\{\mathrm{MOS}_a\}_{a\in\mathcal{A}}$. $\mathcal{A}$ is the collection of three perspectives. Fig. 4 illustrates two examples of generated videos with small (*shaking hands*) and large (*riding bike*) movements. In Fig. 1(c), we visualize the 3D scatter map of human-annotated *subject quality*,

Table 4: **Effects of perspectives.** The correlation between different perspectives for all 9,180 videos in **GAIA**.

| Metric | $\mathrm{MOS}_s \to \mathrm{MOS}_c$ | $\mathrm{MOS}_s \to \mathrm{MOS}_i$ | $\mathrm{MOS}_c \to \mathrm{MOS}_i$ |
|---|---|---|---|
| Spearman's $\rho$ | 0.863 | 0.866 | 0.931 |
| Kendall's $\tau$ | 0.704 | 0.703 | 0.791 |

*action completeness*, and *action-scene interaction* scores in the GAIA dataset and examine three extreme cases, where two dimensions are most differently or consistently (noted in *purple circles*). In general, the generated videos receive lower-than-average human ratings ($\mu_s = 35.48$, $\mu_c = 33.81$, $\mu_i = 30.25$) on three perspectives, suggesting the inferior performance of existing models to produce artifact-free videos with coherent actions. From Tab. 4, we notice a significantly higher correlation between $\mathrm{MOS}_c$ and $\mathrm{MOS}_i$ (*0.931* Spearman's $\rho$, *0.791* Kendall's $\tau$) than other pairs, indicating that

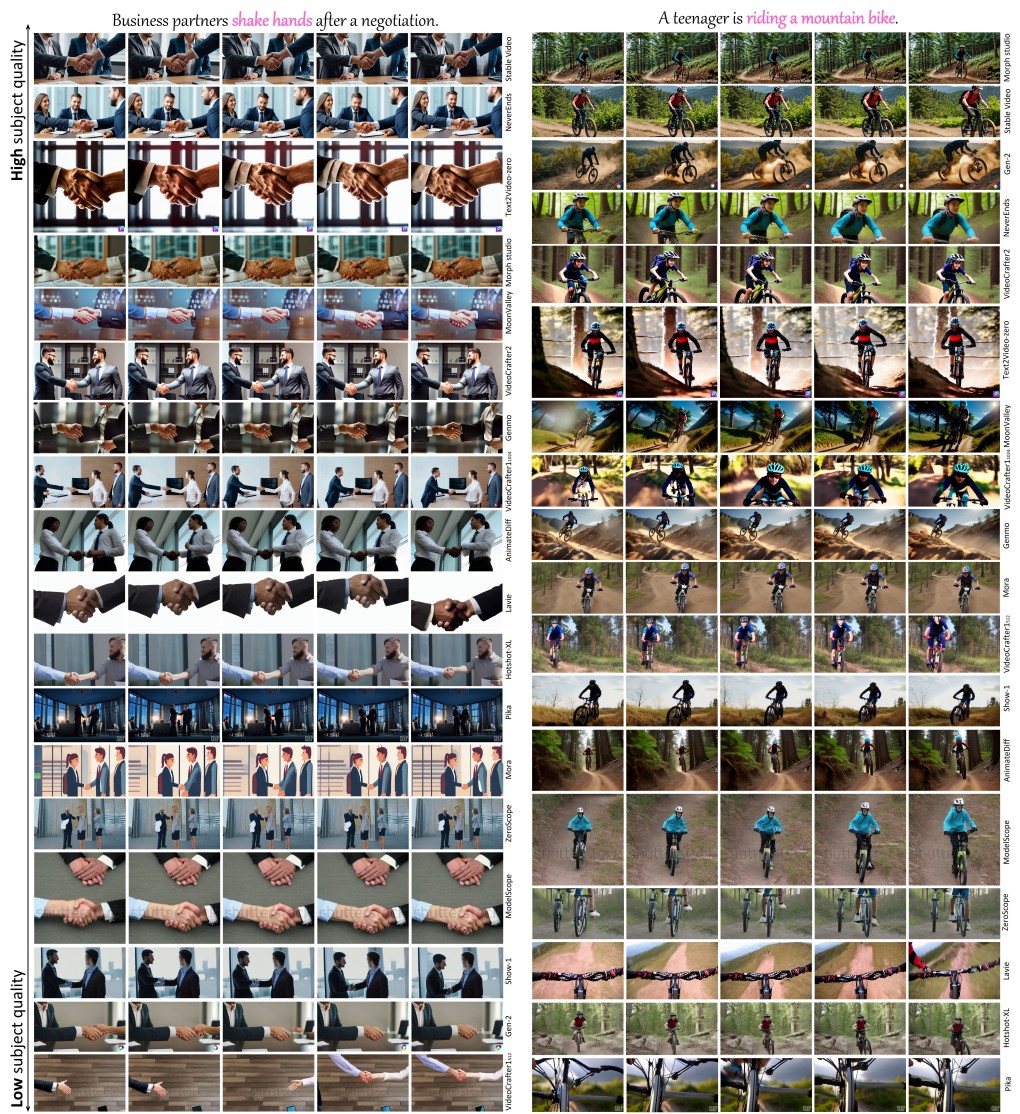

Figure 4: **Visualization of generated videos:** Sort by subject quality from highest to lowest. The action keyword (relatively **small** (*left*) and **large** (*right*) movement) is highlighted in **pink**.

action completeness is a great premise of its rich interaction with the scene context, which further demonstrates the *syllogism*-based action evaluation strategy. More results are in Sec. B.2.1.

**Model-wise Comparison.** As illustrated in Fig. 3 and Fig. 6, the commercial T2V models generally perform better than models from lab studies in three evaluated dimensions. Most models exhibit left-skewed MOS distribution in all three dimensions. Among them, VideoCrafter2 [16] and Morph Studio [4] are basically the best models in their respective fields (see Fig. 13 in the Appendix for detailed ranking in all dimensions). Additionally, we can observe a trend of increasing performance year by year, from the Text2Video-zero [53] and ModelScope [106] released in March 2023 to the VideoCrafter2 [16] in early 2024. Nevertheless, most models prove decent proficiency on one single dimension, *i.e.*, better subject quality than action completeness and action-scene interaction, which exposes the defects of existing models in producing temporal coherent and complete actions. Surprisingly, the newly proposed Mora [118] significantly underperforms other models in all three perspectives, we speculate that it is limited by the core dependency model in its demo code, stable video diffusion (SVD), an earlier image-to-video model. Furthermore, comparing the two resolution versions of VideoCrafter1 ($512 \times 320$ and $1024 \times 576$) [15], as well as the commercial models and the lab studies (with an average resolution of $596 \times 378$ and $1385 \times 835$), we can conclude that higher resolution plays an important role in improving action recognizability, resulting in advancements in

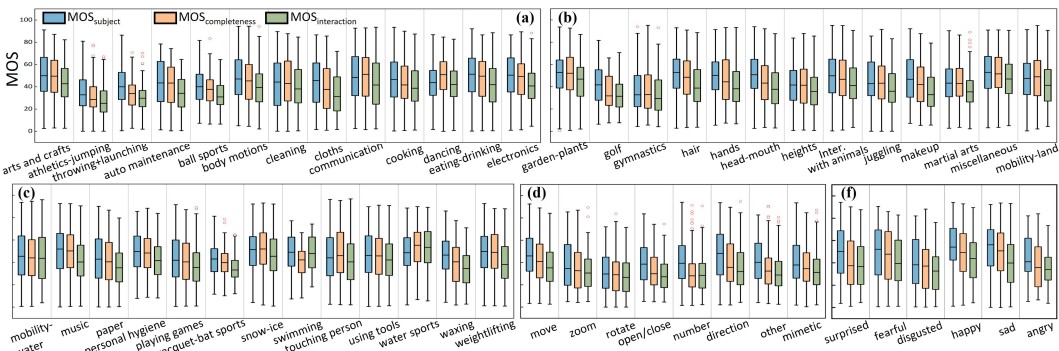

Figure 5: **Box plots of** $MOS_s$**,** $MOS_c$**, and** $MOS_i$ **across action categories.** (a), (b), and (c) show whole-body actions. (d) and (f) show hand and facial actions. For each box, median is the central box, and the edges of the box represent the 25th and 75th percentiles, while red circles denote outliers.

the subject quality and action completeness. A similar conclusion applies to the performance gains as the frame rate increases.

**Class-wise Comparison.** We investigate the MOS distribution across action categories via box plots, as presented in Fig. 5. It can be observed that the $MOS_s$, $MOS_c$, and $MOS_i$ of complex actions such as *jumping/throwing* and *racquet-bat* are lower than actions with small movements (*e.g.*, *communication*, *touching person*, and *using tools*) ($p < 0.01$, Two-side T-test), indicating that existing T2V models struggle to render actions with drastic motion changes, where atypical body postures are more easily involved. Additionally, when it comes to the local hand action categories, the actions contain subtle movements, *e.g.*, *rotate or move fingers/palm*, or *numeral representation* receive significantly lower MOSs than others, showing the inferior capacity of generating fine-grained actions. Specifically, the frequency of outliers in Fig. 5 reflects the response variance of evaluated models under specific action word conditions, which further supports the above viewpoints. *Beyond the above observations, we further analyze the diversity of contents in GAIA (see Sec. B.2).*

## 4 Diagnosis of Automatic Evaluation Metrics

### 4.1 Experimental Setup

To evaluate the performance of conventional AQA methods, we choose four approaches, *i.e.*, USDL [98], ACTION-NET [119], CoRe [117], and TSA [115] for comparison. We also select six action-related metrics from recent T2V benchmarks (VBench [51] and EvalCrafter [67]) as comparisons. Additionally, we include seven representative VQA methods (TLVQM [56], VIDEVAL [102], VSFA [60], BVQA [59], SimpleVQA [97], FAST-VQA [112], and DOVER [113]) to reveal the potential relation between action quality and video quality. We further investigate the performance of video-text alignment metrics, since a high-quality action should be consistent with its target prompt. Seven metrics including four variants of CLIPScore [44] and three vision-language model (VLM)-based metrics, which replace CLIP with more advanced VLMs (BLIP [61], LLaVA-v1.5-7B [65], and InternLM-XComposer2-VL [29]) are evaluated. SRCC and PLCC are adopted as criteria to evaluate the performance of these models. *More implementation details can be found in Sec. C.*

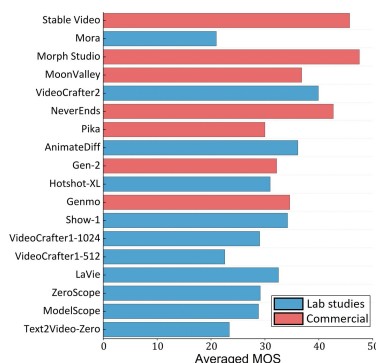

Figure 6: Comparison of T2V models regarding the averaged MOS in three dimensions. We sorted them bottom-up by their release dates.

### 4.2 Main Results and Analysis

**Do conventional AQA methods still work?** As shown in Tab. 5, all AQA methods perform poorly with an average SRCC of $0.4367$, $0.4722$, and $0.4664$ in terms of subject quality, action completeness, and action-scene interaction, respectively. Specifically, USDL takes a manually defined Gaussian

Table 5: **Performance benchmark on GAIA.** *All-Combined* indicates that we sum the MOS of three dimensions and rescale it to [0, 100] as the overall action quality score. ♠, ♣, ◇, and ♡ denote the evaluated conventional AQA method, action-related metrics, VQA methods, and video-text alignment metrics, respectively. All experiments for AQA and VQA methods are retrained on each dimension under 10 random train-test splits at a ratio of 8:2.

| Dimension Methods / Metrics | Pre-training/ Initialization | Subject | | Completeness | | Interaction | | *All-Combined* | |
|---|---|---|---|---|---|---|---|---|---|
| | | SRCC↑ | PLCC↑ | SRCC↑ | PLCC↑ | SRCC↑ | PLCC↑ | SRCC↑ | PLCC↑ |
| ♠USDL (CVPR'20) [98] | | 0.4197 | 0.4203 | 0.4365 | 0.4517 | 0.4289 | 0.4434 | 0.4223 | 0.4321 |
| ♠ACTION-NET (ACM MM'20) [119] | Kinetics [14] | **0.4533** | **0.4612** | 0.4722 | 0.4765 | 0.4703 | 0.4829 | 0.4587 | 0.4592 |
| ♠CoRe (ICCV'21) [117] | | 0.4301 | 0.4343 | 0.4538 | 0.4577 | 0.4521 | 0.4514 | 0.4437 | 0.4415 |
| ♠TSA (CVPR'22) [115] | | 0.4435 | 0.4477 | **0.4963** | **0.4981** | **0.4941** | **0.4953** | **0.4861** | **0.4823** |
| ♣Subject Consistency [51] | DINO [12] | 0.2447 | 0.2362 | 0.2116 | 0.2056 | 0.2034 | 0.1912 | 0.2289 | 0.2273 |
| ♣Motion Smoothness [51] | AMT [63] | 0.2402 | 0.1913 | 0.1474 | 0.1625 | 0.1741 | 0.1693 | 0.1957 | 0.1813 |
| ♣Dynamic Degree [51] | RAFT [99] | 0.1285 | 0.0831 | 0.0903 | 0.0682 | 0.1141 | 0.0758 | 0.1162 | 0.0787 |
| ♣Human Action [51] | UMT [62] | **0.2453** | **0.2369** | **0.2895** | **0.2812** | **0.2861** | **0.2743** | **0.2831** | **0.2741** |
| ♣Action-Score [67] | VideoMAE V2 [107] | 0.2023 | 0.1823 | 0.2867 | 0.2623 | 0.2689 | 0.2432 | 0.2600 | 0.2377 |
| ♣Flow-Score [67] | RAFT [99] | 0.1471 | 0.1541 | 0.0816 | 0.1273 | 0.1041 | 0.1309 | 0.1166 | 0.1430 |
| ◇TLVQM (TIP'19) [56] | NA (*handcraft*) | 0.5037 | 0.5137 | 0.4127 | 0.4158 | 0.4079 | 0.4093 | 0.4655 | 0.4783 |
| ◇VIDEVAL (TIP'21) [102] | NA (*handcraft*) | 0.5237 | 0.5446 | 0.4283 | 0.4375 | 0.4121 | 0.4234 | 0.4684 | 0.4801 |
| ◇VSFA (ACM MM'19) [60] | *None* | 0.5594 | 0.5762 | 0.4940 | 0.5017 | 0.4709 | 0.4811 | 0.5085 | 0.5215 |
| ◇BVQA (TCSVT'22) [59] | *fused* [25, 36, 14, 50, 33] | 0.5702 | 0.5888 | 0.4876 | 0.4946 | 0.4761 | 0.4825 | 0.5201 | 0.5289 |
| ◇SimpleVQA (ACM MM'22) [97] | Kinetics [14] | 0.5920 | 0.5974 | 0.4981 | 0.5078 | 0.4843 | 0.4971 | 0.5219 | 0.5322 |
| ◇FAST-VQA (ECCV'22) [112] | Kinetics [14] | 0.6015 | 0.6092 | 0.5157 | 0.5215 | 0.5154 | 0.5216 | 0.5276 | 0.5475 |
| ◇DOVER (ICCV'23) [113] | LSVQ [116] | **0.6173** | **0.6301** | 0.5198 | 0.5323 | 0.5164 | 0.5278 | **0.5335** | **0.5502** |
| ♡CLIPScore (ViT-B/16) [44] | OpenAI-400M [81] | 0.3360 | 0.3314 | 0.3841 | 0.3777 | 0.3753 | 0.3632 | 0.3777 | 0.3711 |
| ♡CLIPScore (ViT-B/32) [44] | OpenAI-400M [81] | 0.3398 | 0.3330 | 0.3944 | 0.3871 | 0.3875 | 0.3821 | 0.3815 | 0.3826 |
| ♡- - *same as the above* - - | LAION-2B [88] | 0.3179 | 0.3101 | 0.3551 | 0.3511 | 0.3504 | 0.3380 | 0.3531 | 0.3458 |
| ♡CLIPScore (ViT-L/14) [44] | OpenAI-400M [81] | 0.3211 | 0.3156 | 0.3657 | 0.3574 | 0.3585 | 0.3426 | 0.3601 | 0.3515 |
| ♡BLIPScore [61] | COCO [64] | 0.3453 | 0.3386 | 0.4174 | 0.4082 | 0.4044 | 0.3994 | 0.4118 | 0.4054 |
| ♡LLaVAScore [65] | LLaVA-PT [26] | 0.3484 | 0.3436 | 0.4189 | 0.4133 | 0.4077 | 0.4025 | 0.4124 | 0.4086 |
| ♡InternLMScore [29] | *fused* [64, 17, 10, 91, 89] | **0.3678** | **0.3642** | **0.4324** | **0.4257** | **0.4301** | **0.4227** | **0.4314** | **0.4246** |

distribution as the learning objective to address the uncertainty during the human assessment process, which, on the contrary, exacerbates the prediction inaccuracy. Benefiting from the dynamic-static hybrid stream, ACTION-NET can capture the body postures at specific moments during an action process and thus performs marginally better than the rest models in terms of subject quality. For CoRe and TSA, the input requirement is a pairwise query and exemplary video, which is not exactly applicable to AIGVs, since the same action from different models can vary significantly from generation quality to content scenarios, failing the contrastive regression strategy [119, 115]. Most importantly, plagued by the generation quality of AIGV itself, it is difficult for those commonly used inflated 3D ConvNets (I3D) backbone to learn normative action features as in Kinetics [14]. In general, existing AQA methods focus mainly on assessing actions in a similar environment, where the differences between videos are subtle, which is in accord with its goal (*most for specific tasks rather than a generic AQA*).

**Which action-related metric performs better?** As reported in the second part of Tab. 5, all action-related metrics selected from existing benchmarks achieve extremely low correlation in the GAIA dataset with the best scores of 0.2453, 0.2895, and 0.2861 in subject quality, action completeness, and action-scene interaction. Among them, the "Human Action" from VBench [51] and the "Action-Score" from EvalCrafter [67] adopt a similar approach that utilizes the action classification accuracy to quantify the action quality. Their incapability can be attributed to 1) the used recognition model, VideoMAE V2 [107] and UMT [62], are pre-trained only on Kinetics 400 action classes [52] while our GAIA encompasses much broader action types; 2) based on the premise that action subject is clearly visible and temporally consistent, a condition that is challenging to fulfill in the majority of existing AIGVs. Using optical flow-based metrics, "Dynamic Degree" and "Flow-Score", to measure the movement of actions fails since the motion amplitude of different actions varies. "Motion Smoothness" is proposed to evaluate whether the motion in AIGVs follows the physical law of the real world based on the frame interpolation theory [51]. However, it is not conducive to videos with a low frame rate and cannot justify the rationality of the generated action result such as *badminton ball flying against gravity*. As for the "Subjective Consistency" metric, there is a potential for misapplication in assessing the quality of the subject, since variability in subject posture throughout the action can easily lead to inter-frame subject inconsistencies. Consolidating the above experimental results, we can conclude that current action evaluation metrics fall short of providing reliable action assessments, necessitating a concerted effort to address these issues for the emerging AIGVs.

**Comparison to the VQA methods.** Considering the intrinsic correlation of action quality on the content quality of videos, we select seven representative VQA methods to validate whether VQA

approaches are applicable for AQA tasks in AI-generated scenarios, as shown in the third part of Tab. 5. We can observe that VQA methods surpass all AQA methods and action-related metrics by a large margin (on average 25.04% and 131.1% better than their respective best methods in terms of SRCC) in the subject quality dimension, while deep learning-based VQA methods perform better than traditional VQA methods (TLVQM and VIDEVAL) that rely on handcraft features. Notably, all VQA methods exhibit a relatively superior capacity to evaluate the subject quality than assessing the action completeness and action-scene interaction, indicating a potential emphasis on low-level technical distortions such as noises, sharpness, blur, and artifacts within the current VQA frameworks, which may not fully encapsulate the temporal-level normativity and interactive facets of action content. Such a conclusion is also supported by evidence from being equipped with different quality-aware initializations, as BVQA and DOVER are pre-trained with spatial distortion-dominated datasets [25, 36, 50, 33, 116]. Moreover, BVQA and SimpleVQA leverage the SlowFast model [34] as their motion feature extractor. This model has demonstrated effectiveness in various action recognition tasks due to its dual pathway design, which captures both spatial semantics and motion information parallelly. However, it encounters problems when applied to AIGVs, primarily because of the limited frames. Another plausible explanation for these subpar performances is the pure regression-based prediction strategy that lacks consideration of textual information, as the same MOS for different actions could lead to a large visual discrepancy.

**Evaluation on video-text alignment metrics.** We further evaluate the performance of video-text alignment metrics in measuring action quality considering their capacity in cross-modality feature mapping. Specifically, we compute the cosine similarity between the image embedding and the action prompt embedding to record a deviation degree between the sketch of the content and target action semantics. As listed in Tab. 5, the widely used CLIPScore achieves a weak correlation with human opinion, especially in the subject quality dimension, while performing relatively better with respect to action completeness and action-scene interaction dimensions. We conjecture that this is because such alignment-based metrics are intrinsically sensitive to high-level vision information (*action semantics*) rather than low-level generative flaws (*e.g., blur, noise, textures*). Meanwhile, we see a decent performance gain on evaluated dimensions (+8.2%, +9.6%, +10.9%, +13.1% in terms of SRCC) when replacing CLIP with a more powerful VLM, such as InternLM-XComposer2-VL, showing an underlying possibility of building more accurate AQA metrics as VLMs evolve. We also conduct a T-test with a 95% confidence level to assess the statistical significance of the performance difference between any two methods (Tab. 9 and Fig. 15). More results are discussed in Sec. D.

## 5   Conclusion

Assessing action quality in AI-generated videos is a critical topic since it is an intuitive manifestation of the model generation ability and an imperative factor influencing the viewing experience of a video that requires data beyond the currently available prompt and video pairs datasets. We present GAIA, a well-curated generic AI-generated action dataset comprising 9,180 videos generated from 18 popular T2V models with 971,244 human annotations collected. We use it to evaluate the action generation ability of existing T2V models and benchmark the performance of current AQA and VQA methods. Our analysis characterizes the distinctness, variation, and capacity evolution of existing T2V models while revealing the inferiority of traditional AQA and VQA algorithms in providing subjectively consistent action quality assessments for AI-generated videos. We hope that GAIA will facilitate the development of accurate AQA algorithms for AIGVs while elucidating the factors to which humans are sensitive during action perception.

**Limitations and Societal Impact.** First, the videos in our dataset are limited in scope concerning subject types and styles, which constrains its applicability. The current synthetic actions are relatively simple as opposed to the complicated motions in real life. Second, videos in our dataset are generated with limited resolutions, frame rates, or lengths due to the imbalance between industry and academia, which could be further refined as the T2V model evolves. Third, different from prior work [78, 119, 66, 76, 115, 122], the action annotations in our dataset were collected based on a causal reasoning syllogism, which stands in stark contrast to the conventional practice of collecting a single quality score. Investigations of such a strategy on AQA would be a fruitful avenue for follow-up work. We anticipate that this work will lead to improved action quality in AI-generated videos, promoting the development of objective AQA metrics in generation domains and the understanding of human action perception mechanisms. Besides, this can make models pre-trained on this dataset less biased in assessing incomplete actions and irrational actions that easily appear in AI-generated scenarios.

# 6    Acknowledgements

This work was supported in part by the China Postdoctoral Science Foundation under Grant Number 2023TQ0212 and 2023M742298, in part by the Postdoctoral Fellowship Program of CPSF under Grant Number GZC20231618, in part by the Shanghai Pujiang Program under Grant 22PJ1407400, and in part by the National Natural Science Foundation of China under Grant 62271312, 62301316, 62101325 and 62101326.

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

Table 6: **Comparison of existing T2V benchmarks.** [†]We only report the number of dimensions with user opinion alignments. GAIA is a generic AI-generated action dataset that focuses more on the action quality in videos while owning more diverse model types and human annotations.

| Benchmark | Videos | Prompts | Models | [†]Dimension | Total Ratings | Annotators |
|---|---|---|---|---|---|---|
| Chivileva *et al.* (arxiv2023) [23] | 1,005 | 201 | 5 | 2 | 48,240 | 24 |
| FETV (NeurIPS2023) [69] | 2,476 | 619 | 4 | 4 | 28,116 | 3 |
| EvalCrafter (CVPR2024) [67] | 3,500 | 500 | 7 | 5 | − | 3 |
| VBench (CVPR2024) [51] | 6,984 | 1,746 | 4 | 16 | − | − |
| T2VQA-DB (arxiv2024) [57] | 10,000 | 1,000 | 9 | 2 | − | 27 |
| **GAIA (Ours)** | 9,180 | 510 | 18 | 3 | 971,224 | 54 |

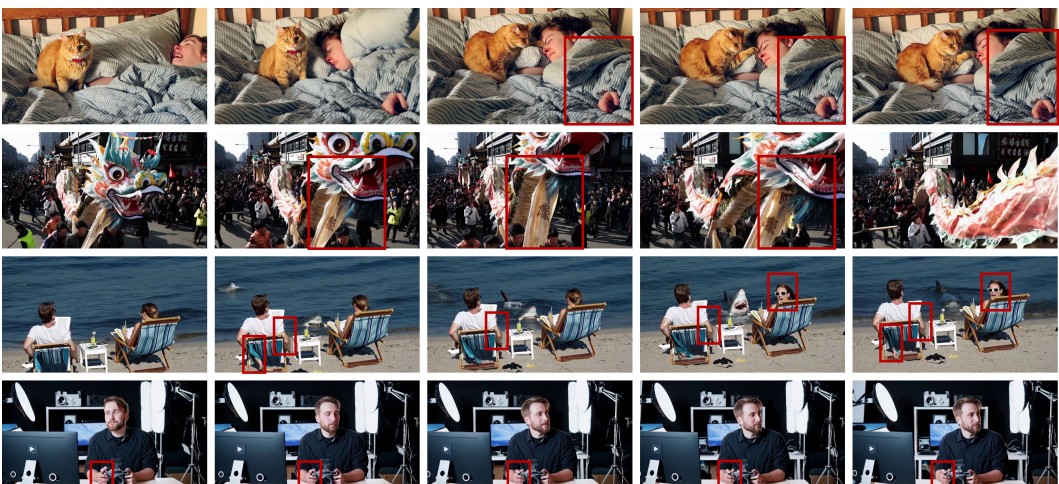

Figure 7: **Examples of action-related abnormal content in Sora generated videos.** 1st row: A cat owner rolls over in bed with an *unnatural body position*; 2nd row: The head of Chinese dragon is raised *without a holding point*; 3rd row: A woman frightened by a shark turns her head at an *incredible angle* and a man reading a book with *duplicate hands*. 4th row: A man holding his camera with *six fingers*. These video clips suffer from problems of action subject quality and action-scene interaction. The red rectangles indicate areas within individual frames where the action appears unnatural.

# Appendix

## A Ethical Discussions

### A.1 Ethical Discussions of Our Research

Our work holds the potential for significant social impact, both positive and negative. We anticipate that this work will raise consideration of human perception and understanding in AI-generated actions to better understand generative models and enable more predictable behavior. Currently, the human preference and perceptual sensitivity to the quality of action along the whole action process still remains as an open problem. This work also provides significant guidance on how to optimize video generation models to produce videos with more pleasant actions. Meanwhile, we acknowledge that this study could raise some safety and ethical concerns. One challenging aspect of text-to-video models is the generation of NSFW content (such as violent and pornographic contents), which may be offensive or inappropriate for some viewers and can potentially foster illegal transactions. Although some video generation platforms like MoonValley, Morph and Stable Video have built-in safety filters that detect prompts with NSFW contents, they can still be circumvented through prompt engineering [18]. Additionally, AI video generation technology can be exploited by criminals for fake impersonations and identity theft. Our study also highlighted that some AI-generated videos can convincingly mimic individual's facial expressions and actions, thereby posing a latent threat to public safety and eroding public trust in social media.

We further discuss how our work can be applied to benefit the community. **Firstly**, the main motivation of our work is that the action-related contents highly affect the video viewing experience, especially in this era where AI-generated models are prevalent, yet current video generation models inevitably suffer from subpar action quality, visual artifacts, and temporal inconsistencies within the generated actions. The existing action quality assessment (AQA) research is highly domain-specific, leading to a relatively poor generalization ability across tasks. Due to the domain gap between real videos and AI-generated videos (AIGVs) as well as the difference in task orientation, previous AQA methods underperform in AI generation scenarios. In terms of data sources, existing AQA studies collect the quality scores directly from judges or minority groups (Tab. 1), which is applicable in professional events but can introduce bias in studying the group preference. The mechanisms by which humans assess the quality of actions and the underlying influences are unknown. In this work, we find that the actions from mainstream T2V models are still subpar in subject quality, action completeness, and action-scene interaction perspectives (even Sora [73] shown in Fig. 7), while neither existing AQA algorithms nor video quality assessment (VQA) methods are suitable for evaluating action quality in AIGVs. Our findings underline the necessity of developing reliable automatic AQA metrics for AIGVs while taking the first step to evaluate the action quality in AIGVs through a causal reasoning manner, which also provides valuable insights for the community in refining video generation models. **Secondly**, despite the action quality, a common line of works tries to evaluate AI-generated videos from traditional spatial quality (*e.g.*, *fidelity*, *blur*, *brightness*, and *aesthetic*) and temporal quality (*e.g.*, *light change*, *background consistency*, *warping error*, and *motion quality*) perspectives [51, 23, 69, 67, 57]. Tab. 6 gives a brief comparison of existing T2V benchmarks. While these lines of work serve a general purpose, their action-related metrics were simply adapted from previous action representation strategies used in real world, which is less effective and exhibits inconsistency with human perception in AI-generated scenarios. Our work helps to build a more reasonable definition of action quality in AIGVs. **Thirdly**, evaluating action quality in a causal reasoning way offers a promising way to understand human action perception and test the performance of T2V models, thus pointing the path for the future improvement of video generation models.

## A.2 Ethical Discussions of Data Collection

We detail the ethical concerns that might arise in the dataset collection. All participants in subjective evaluation are clearly informed of the contents in our experiments. Specifically, we addressed the ethical challenges by obtaining from each subject depicted in the dataset a signed and informed agreement that they agreed their subjective ratings to be used for non-commercial research, making it equipped with such legal and ethical characteristics. The experiments do not contain any visually inappropriate content or NSFW content (both *textual* and *visual*) since we applied rigorous manual review during the action generation stage. Considering the large number of evaluated videos, we divided 9,180 videos into 31 sessions. Fig. 8 exhibits the user interface for collecting subjective opinions. Each participant was compensated $12 for each session according to the current ethical standard [92, 74]. It took over a month to complete the whole experiment, where each participant contributed an average of 80.6 hours to attend this experiment. To ensure participants' anonymity, we numbered 54 participants according to the order of participation into $P_1 \dots P_{54}$ and performed a questionnaire survey about their sex, age, and whether they had used AI generation tools, which are not considered as person identifiable information. Note that we do not disclose this information in our dataset, which is used only for reporting participants' statistics. The **GAIA** dataset is released under the **CC BY 4.0** license, which includes all associated AIGVs and their corresponding action prompts.

## B More Details of GAIA Dataset

We listed the URL of the adopted text-to-video models in Tab. 7 and detailed the category of each action keyword in our GAIA dataset in Tab. 12, Tab. 13, Tab. 14, and Tab. 15.

### B.1 Detailed Information of Text-to-Video Models

**Text2Video-Zero.** Text2Video-Zero [53] is a zero-shot text-to-video (T2V) synthesis model without any further fine-tuning or optimization, which introduces motion dynamics between the latent codes and cross-frame attention mechanism to keep the global scene time consistent. We adopt its official

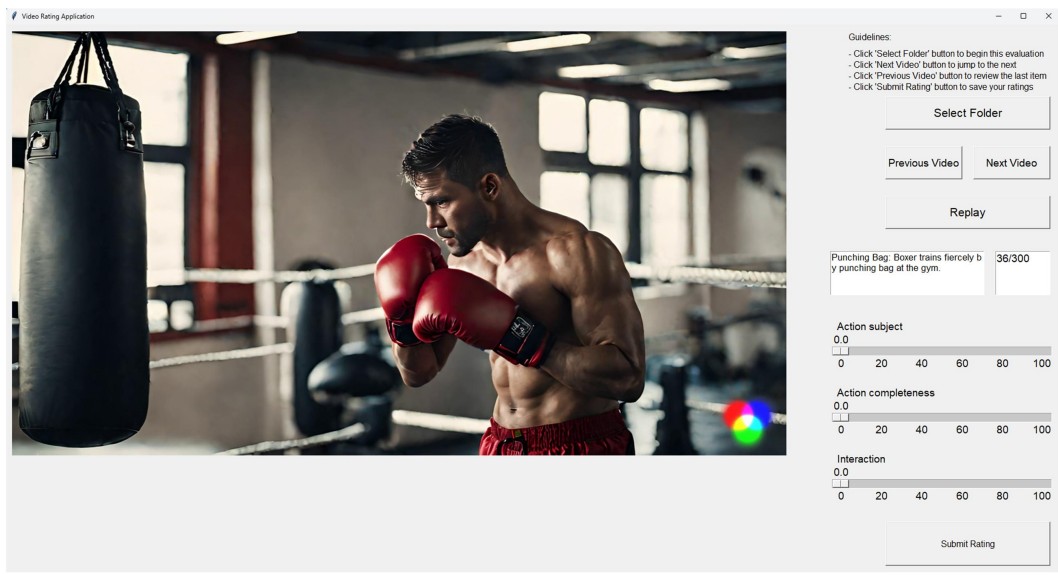

Figure 8: **Screenshot of the rating interface for human evaluation.** Participants are instructed to rate three action-related dimensions of AI-generated videos, *i.e.*, *subject quality*, *action completeness*, and *action-scene interaction*, based on the given action keyword and prompt.

Table 7: URLs for the adopted text-to-video models.

| Methods | URL |
|---|---|
| Text2Video-Zero [53] | https://github.com/Picsart-AI-Research/Text2Video-Zero |
| ModelScope [106] | https://modelscope.cn/models/iic/text-to-video-synthesis/summary |
| ZeroScope [8] | https://huggingface.co/cerspense/zeroscope_v2_576w |
| LaVie [110] | https://github.com/Vchitect/LaVie |
| Show-1 [120] | https://github.com/showlab/Show-1 |
| Hotshot-XL [71] | https://github.com/hotshotco/Hotshot-XL |
| AnimateDiff [41] | https://github.com/guoyww/AnimateDiff |
| VideoCrafter1-512 [15] | https://github.com/AILab-CVC/VideoCrafter |
| VideoCrafter1-1024 [15] | https://github.com/AILab-CVC/VideoCrafter |
| VideoCrafter2 [16] | https://github.com/AILab-CVC/VideoCrafter |
| Mora [118] | https://github.com/lichao-sun/Mora |
| Gen-2 [1] | https://research.runwayml.com/gen2 |
| Genmo [2] | https://www.genmo.ai |
| Pika [6] | https://pika.art/home |
| NeverEnds [5] | https://neverends.life |
| MoonValley [3] | https://moonvalley.ai |
| Morph Studio [4] | https://www.morphstudio.com |
| Stable Video [7] | https://www.stablevideo.com/welcome |

code with default parameters (`<motion_field_strength_x&y=12>`, $t0 = 44$, $t1 = 47$) and sample 8 frames of size $512{\times}512$ at 4 frames per second (FPS).

**ModelScope.** ModelScope [106] is a multi-stage diffusion-based T2V generation model. We use the official inference code and sample 15 frames of size $256{\times}256$ at 8 FPS.

**ZeroScope.** ZeroScope [8] is a Modelscope-based [106] video model optimized for producing 16:9 compositions. We use the official inference code and sample 24 frames of size $576{\times}320$ at 8 FPS. The number of inference steps is set to 40.

**LaVie.** LaVie [110] is an integrated video generation framework that operates on cascaded video latent diffusion models. For each prompt, we use the base T2V model and sample 16 frames of size $512{\times}320$ at 8 FPS. The number of DDPM [47] sampling steps and guidance scale are set as 50 and 7.5, respectively.

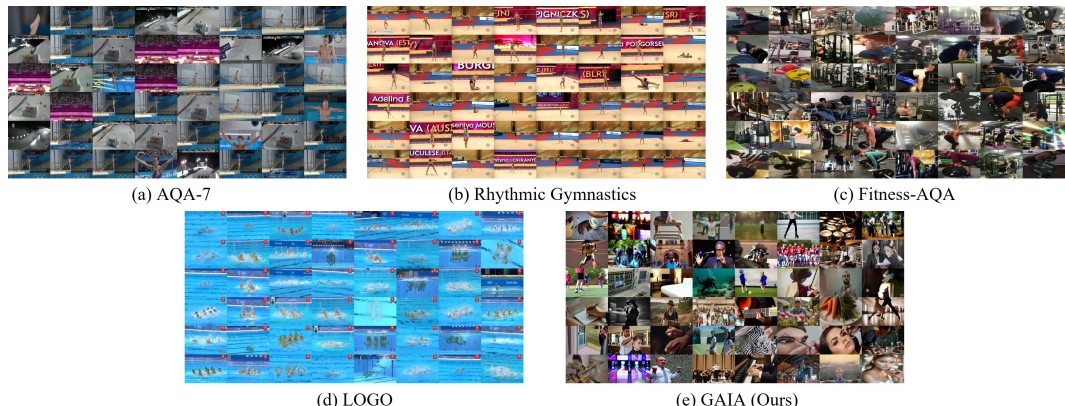

Figure 9: **Sample frames of the video contents contained in five representative AQA datasets:** (a) AQA-7 [77], (b) Rhythmic Gymnastics [119], (c) Fitness-AQA [76], (d) LOGO [122], and the proposed (e) GAIA. Compared to other datasets that include only a single class of actions happening in specific scenes, GAIA comprises more diverse actions generated by text-to-video models.

**Show-1.** Show-1 [120] is a hybrid model which marries pixel-based and latent-based T2V diffusion models. It first produces a set of low-resolution key frames with strong text-video correlation and then employs frame interpolation and spatial upscaling to generate high-quality videos. We use the official inference code with parameters of `<num_base_steps=75, num_interpolation_steps=75, num_sr1_steps=125, num_sr2_steps=50>` and sample 29 frames of size 576×320 at 8 FPS.

**Hotshot-XL.** Hotshot-XL [71] is a text-to-gif model trained to work alongside Stable Diffusion XL[1]. We change the output format from `GIF` to `MP4` and sample 8 frames of size 672×384 at 8 FPS.

**AnimateDiff.** AnimateDiff [41] is a practical framework for animating personalized text-to-image models, which enables a pre-trained motion module to adapt to new motion patterns without requiring model-specific tuning. We use the general T2V version of `AnimateDiff_v3` with default parameters and sample 16 frames of size 384×256 at 8 FPS.

**VideoCrafter.** VideoCrafter is a video generation and editing toolbox. We utilize the generic T2V generation model: VideoCrafter1 [15] and VideoCrafter2 [16]. For VideoCrafter1, we sample 16 frames of size 512×320 and 1024×576 at 8 FPS, according to its default settings. For VideoCrafter2, we sample 16 frames of size 512×320 at 8 FPS.

**Mora.** Mora [118] is a recent multi-agent framework that incorporates several advanced visual AI agents to achieve generalist video generation, which mainly consists of text-to-image, image refine and image-to-video procedures. We use the officially open-sourced demo that takes stable diffusion[2] as inference pipeline. 100 frames of size 1024×576 at 25 FPS are sampled for each prompt.

**Gen-2.** Gen-2 [1] is a multimodal AI system, introduced by Runway AI, Inc., which can generate novel videos with text, images or video clips. We collect 96 frames of size 1408×768 at 24 FPS for each prompt. The intensity of motion is set to 5.

**Genmo.** Genmo [2] is a high-quality video generation platform. We generate 60 frames of size ≤2048×1536 at 15 FPS for each prompt. The motion parameter is set to 70%.

**Pika, NeverEnds, MoonValley, Morph Studio.** Pika [6], NeverEnds [5], MoonValley [3], and Morph Studio [4] are recent popular online video generation application. We use the T2V mode of these application via command in Discord[3]. For Pika, we generate 72 frames of size 1088×640 at 24 FPS for each prompt. For NeverEnds, we generate 30 frames of size 1024×576 at 10 FPS for each prompt. For MoonValley, we set `<style='realism', duration='medium'>` and generate 187 frames of size 1184×672 at 50 FPS for each prompt. For Morph Studio, we generate generate 72

---

[1]https://huggingface.co/hotshotco/SDXL-512
[2]https://huggingface.co/stabilityai
[3]https://discord.com

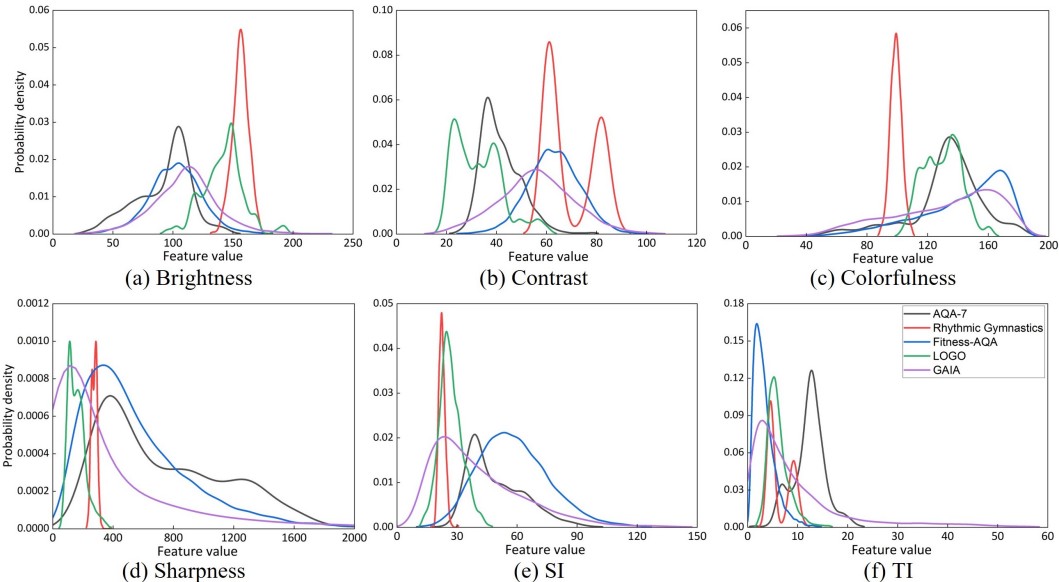

Figure 10: **Feature distribution comparisons among five AQA datasets:** AQA-7 [77], Rhythmic Gymnastics [119], Fitness-AQA [76], LOGO [122], and the proposed GAIA.

frames of size $1920\times1080$ at 24 FPS for each prompt. Limited by the response speed and the number of requests, the overall duration of collecting these videos exceed **200** hours.

**Stable Video.** Stable Video [7] is Stability AI's reference implementation for the latest video models. We use the T2V mode in web application without adding camera motion settings. The inference steps and motion strength are set to 40 and 127, respectively. For each prompt, we obtain 96 frames of size $1024\times576$ at 24 FPS.

## B.2 Quantitative and Qualitative Comparison of Content

Fig. 9 shows some representative snapshots of the source sequences for five representative AQA datasets, respectively. As a way of characterizing the content diversity of the videos in each dataset, we calculate six low-level features including brightness, contrast, colorfulness, sharpness, spatial information (SI), and temporal information (TI) [102], thereby providing a large visual space in which to plot and analyze content diversities of the five AQA datasets. To reasonably reduce the computational overhead, each of these features was computed on every 8th frame, then averaged over frames to obtain an overall feature representation of each content. Here, we denote the feature as $\{F_i\}, i = 1, 2, \ldots, 6$. Fig. 10 shows the fitted kernel distribution of each selected feature. We also plotted convex hulls of paired features in Fig. 11 to show the feature coverage of each dataset. Furthermore, to quantify the coverage and uniformity of these datasets over each feature space, we computed the relative range and uniformity of coverage [111]. Concretely, the relative range is given by:

$$R_i^k = \frac{\max(D_i^k) - \min(D_i^k)}{\max_k(D_i^k)}, \tag{1}$$

where $D_i^k$ denotes the feature distribution of dataset $k$ for a given feature dimension $i$. $\max_k(D_i^k)$ specifies the maximum value for that given dimension across all datasets. The entropy of the B-bin histogram of $D_i^k$ over all sources for each dataset $k$ is calculated to quantify the uniformity of coverage, which stands for how uniformly distributed the videos are in each feature dimension:

$$U_i^k = -\sum_{b=1}^{B} p_b \log_B p_b, \tag{2}$$

where $p_b$ denotes the normalized number of contents in bin $b$ at feature $i$ for dataset $k$. The higher the uniformity (Fig. 12(b)), the more uniform the database is, which together with the relative range (Fig. 12(a)) measures the intra- and inter-dataset differences, respectively.

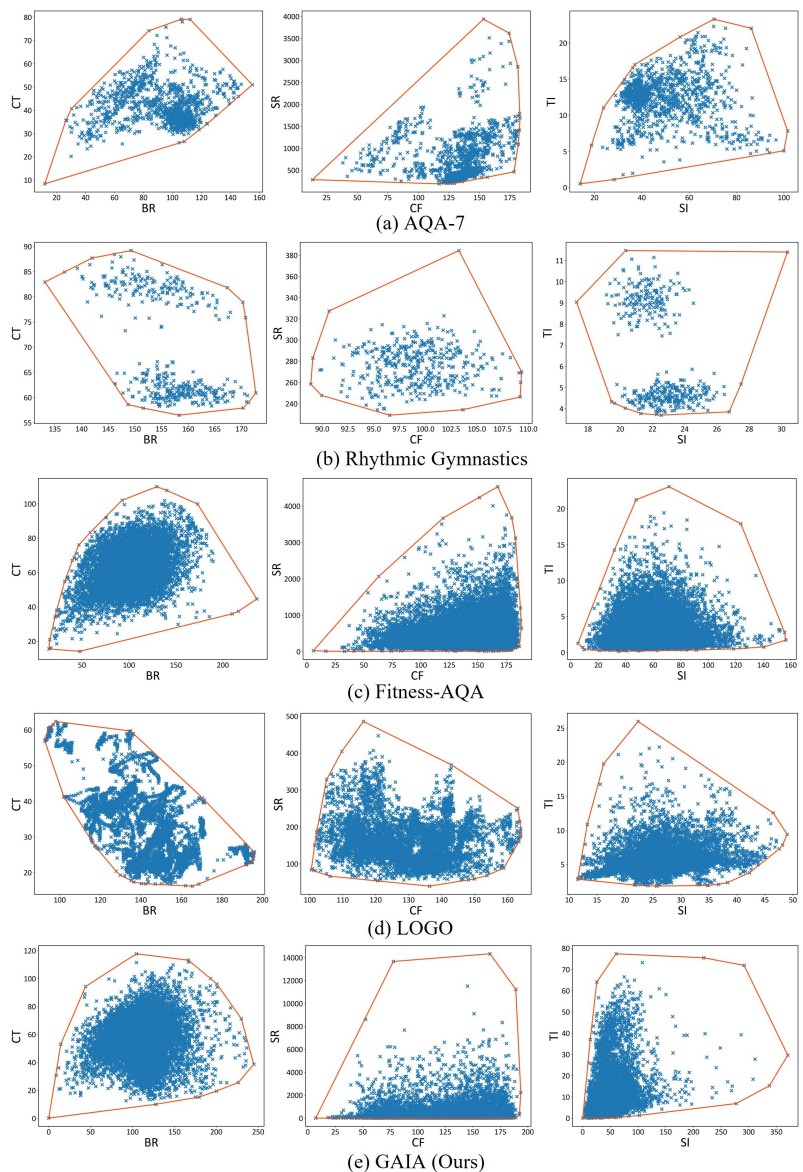

Figure 11: Source content (blue 'x') distribution in paired feature space with corresponding convex hulls (orange boundaries). Left column: BR×CT, middle column: CF×SR, right column: SI×TI.

Given the above plots, we make some observations. As can be seen in Fig. 10 and the corresponding convex hulls in Fig. 11, AQA-7, Rhythmic Gymnastics, and LOGO exhibit a sharply peaked distribution within a narrow range of feature values, indicating the singularity of the action scenes, which is consistent with the snapshot visualized in Fig. 9. On the contrary, our GAIA and Fitness-AQA own a wider range of features and are closer to the normal distribution. Similarly, we can observe from Fig. 12(a) that our GAIA spread most widely in all six dimensions. However, the coverage uniformity of GAIA is significantly lower than the other datasets in terms of sharpness, SI, and TI, which we attribute to the differences in generated models. Compared to datasets collected from real-world action video sources, GAIA is composed of AI-generated videos generated with varied spatial resolution and frame rate settings. Besides, the variety of actions also affects the uniformity of temporal information. The above observations together verify the novelty and variations of AI-generated videos in the proposed GAIA dataset, thus demonstrating its qualification to serve as a generic AQA dataset to facilitate the future development of AQA algorithms.

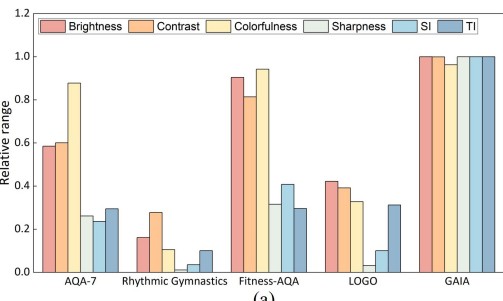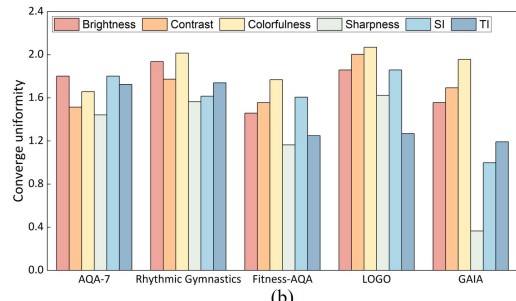

Figure 12: Comparisons of the selected six features calculated on the five AQA datasets: AQA-7 [77], Rhythmic Gymnastics [119], Fitness-AQA [76], LOGO [122], and the proposed GAIA: (a) Relative range $R_i^k$; (b) Coverage uniformity $U_i^k$.

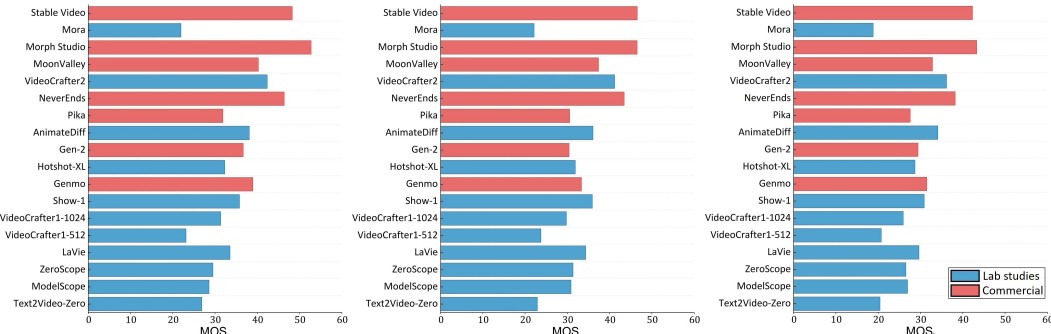

Figure 13: Detailed model-wise comparison in terms of $\mathrm{MOS}_s$, $\mathrm{MOS}_c$, $\mathrm{MOS}_i$.

### B.2.1 More Statistics of GAIA

We provide the scatter plots about MOS against standard deviation (STD), along with the five-parameter polynomial fitting plot (orange line) in Fig. 14. First, there is a relatively linear distribution of STD for all three perspectives with MOS<15, suggesting that humans are more consistent in perceiving poor-quality actions. Similar observations can be found in high MOS scenarios (MOS>90). Second, the trend lines reveal a peak in STD distribution when MOS is in $[20, 40]$, with a steeper decline and increase in the high MOS range (MOS>80) and low MOS range (MOS<30), respectively. We speculate that AI-generated high-quality actions are mostly consistent with people's common sense, whereas medium- and low-quality actions exhibit greater diversity, leading to a more pronounced divergence among individuals. Another plausible explanation is that this is due to the uneven distribution of high and low quality action videos in GAIA. Third, the STD distribution is narrower for the subject quality dimension than for action completeness and action-scene interaction dimensions, indicating that the perception of spatial quality distortion in action is less divergent than the temporal consistency and rationality distortion.

## C Implementation Details

Our experiments were conducted on a computer with Intel Core i9-14900K CPU@3.20GHz, 64GB RAM, and NVIDIA RTX 4090 24GB. Tab. 8 lists the URL of the evaluated baselines. All experiments for AQA and VQA methods are retrained on each evaluated dimension under 10 random train-test splits at a ratio of 8:2.

### C.1 Evaluation Metrics

We adopt the widely used metrics in AQA and VQA literature [22, 78]: Spearman rank-order correlation coefficient (SRCC) and Pearson linear correlation coefficient (PLCC), as our evaluation criteria. SRCC quantifies the extent to which the ranks of two variables are related, which ranges

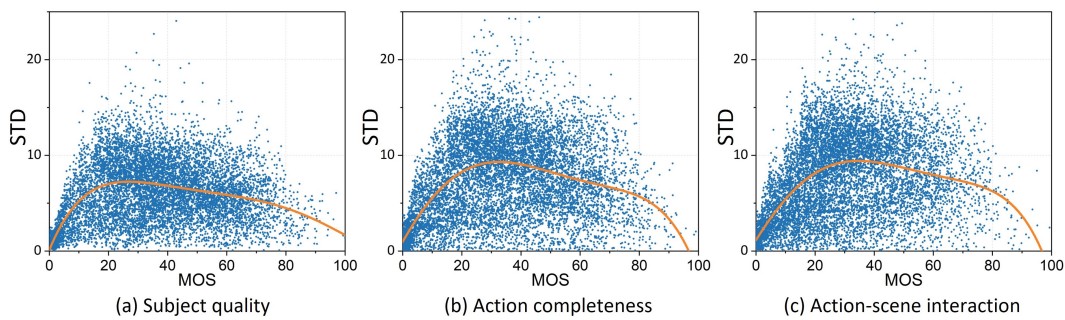

Figure 14: Scatter plots about MOS against its standard deviation (STD) and five-parameter polynomial fitting plots (orange line) of three perspectives of action quality: (a) subject quality, (b) action completeness, and (c) action-scene interaction.

Table 8: URLs for the compared automatic evaluation methods.

| Methods / Metrics | URL |
|---|---|
| USDL (CVPR'20) [98] | https://github.com/nzl-thu/MUSDL |
| ACTION-NET (ACM MM'20) [119] | https://github.com/qinghuannn/ACTION-NET |
| CoRe (ICCV'21) [117] | https://github.com/yuxumin/CoRe |
| TSA (CVPR'22) [115] | https://github.com/xujinglin/FineDiving |
| Subject Consistency [51] | |
| Motion Smoothness [51] | |
| Dynamic Degree [51] | https://github.com/Vchitect/VBench |
| Human Action [51] | |
| Action-Score [67] | |
| Flow-Score [67] | https://github.com/EvalCrafter/EvalCrafter |
| TLVQM (TIP'19) [56] | https://github.com/jarikorhonen/nr-vqa-consumervideo |
| VIDEVAL (TIP'21) [102] | https://github.com/vztu/VIDEVAL |
| VSFA (ACM MM'19) [60] | https://github.com/lidq92/VSFA |
| BVQA (TCSVT'22) [59] | https://github.com/zwx8981/TCSVT-2022-BVQA |
| SimpleVQA (ACM MM'22) [97] | https://github.com/sunwei925/SimpleVQA |
| FAST-VQA (ECCV'22) [112] | https://github.com/VQAssessment/FAST-VQA-and-FasterVQA |
| DOVER (ICCV'23) [113] | https://github.com/VQAssessment/DOVER |
| CLIPScore (ViT-B/16) [44] | |
| CLIPScore (ViT-B/32) [44] | https://github.com/jmhessel/clipscore |
| CLIPScore (ViT-L/14) [44] | |
| BLIPScore [61] | https://github.com/salesforce/BLIP |
| LLaVAScore [65] | https://huggingface.co/llava-hf/llava-1.5-7b-hf |
| InternLMScore [29] | https://huggingface.co/internlm/internlm-xcomposer2-vl-7b |

from -1 to 1. Given $N$ action videos, SRCC is computed as:

$$SRCC = 1 - \frac{6 \sum_{n=1}^{N} (v_n - p_n)^2}{N(N^2 - 1)}, \tag{3}$$

where $v_n$ and $p_n$ denote the rank of the ground truth $y_n$ and the rank of predicted score $\hat{y}_n$ respectively. The higher the SRCC, the higher the monotonic correlation between ground truth and predicted score. Similarly, PLCC measures the linear correlation between predicted scores and ground truth scores, which can be formulated as:

$$PLCC = \frac{\sum_{n=1}^{N} (y_n - \bar{y})(\hat{y}_n - \bar{\hat{y}})}{\sqrt{\sum_{n=1}^{N} (y_n - \bar{y})^2} \sqrt{\sum_{n=1}^{N} (\hat{y}_n - \bar{\hat{y}})^2}}, \tag{4}$$

where $\bar{y}$ and $\bar{\hat{y}}$ are the mean of ground truth and predicted score respectively.

## C.2 Action Quality Assessment Methods

**USDL** [98] is an uncertainty-aware score distribution learning approach for AQA, which regards an action as an instance associated with a score distribution. Considering the varied frame length of videos in GAIA, we do not perform frame segmentation for those with less than 16 frames, but

uniformly divide other videos into ten segments. I3D backbone pre-trained on Kinetics[4] is used for feature extraction. For the final score, since it was a float number, we normalized it as:

$$S^k_{normalized} = \frac{S^k - S^k_{min}}{S^k_{max} - S^k_{min}} \times 100, \tag{5}$$

where $S^k_{min}$ and $S^k_{max}$ are the minimum and maximum score of the $k$-th dimension in GAIA. After that, we produced a Gaussian function with a mean of $S^k_{normalized}$ as in [98]. Other settings are adopted as the official recommendations.

**ACTION-NET** [119] is a hybrid dynamic-static context-aware attention network for AQA in long videos, which not only learns the video dynamic information but also focuses on the static postures of the detected action subjects in specific frames. For the dynamic stream, we sampled 4 frames per second. For the static stream, we sampled the first, middle, and last frames, then applied the same detection algorithm as the author did to crop the region with the detected action subject.

**CoRe** [117] formulates the problem of AQA as regressing the relative scores with reference to another video that has shared attributes such as action category, which utilizes the differences between action videos and guides the model to learn the key hints for assessment. Due to the differences between the categorization strategy of our GAIA and that of the AQA-7 dataset used in the original experiment, we randomly select a video of the same action generated by another T2V model as the exemplar video. We evenly segmented each video clip into 4 snippets, each containing 4 continuous frames. For those videos less than 16 frames long, we applied frame interpolation to satisfy the length requirement.

**TSA** [115] is a temporal segmentation attention module placed after the spatial-temporal visual feature extraction to successively accomplish procedure-aware cross-attention learning. Similar to CoRe, we evenly segmented each video clip into 4 snippets, each containing 4 continuous frames, and then fed them into I3D. Other settings are adopted as the official recommendations.

### C.3 Action-related Metrics

For **Subject Consistency**, **Motion Smoothness**, **Dynamic Degree**, **Human Action**, **Action-Score**, and **Flow-Score** metrics, we directly used their respective implementation code in VBench [51] and EvalCrafter [67] without specific changes.

### C.4 Video Quality Assessment Methods

**TLVQM** [56] is a two-level video quality model, which is based on the idea of computing features in two levels so that low complexity features are computed for the full sequence first, and then high complexity features are extracted from a subset of representative video frames, selected by using the low complexity features. **VIDEVAL** [102] employs a feature selection strategy on top of efficient blind VQA models. We used the official open-sourced codes and transformed the format of videos in GAIA from RGB space to YUV420 for feature extraction.

**VSFA** [60] is an objective no-reference video quality assessment method by integrating two eminent effects of the human visual system, namely, content-dependency and temporal-memory effects into a deep neural network. We directly used the official code without specific changes.

**BVQA** [59] leverages the transferred knowledge from image quality assessment (IQA) databases with authentic distortions and large-scale action recognition with rich motion patterns for better video representation. We used the officially pre-trained model under mixed-database settings [25, 36, 14, 50, 33] and finetuned it on our GAIA for evaluation.

**SimpleVQA** [97] adopts an end-to-end spatial feature extraction network to directly learn the quality-aware spatial feature representation from raw pixels of the video frames and extract the motion features to measure the temporal-related distortions. A pre-trained SlowFast model is used to extract motion features. Specifically, we uniformly sampled 8 frames while rescaling them at a fixed height of 520 as inputs.

**FAST-VQA** [112] proposes a grid mini-patch sampling (GMS) strategy, which allows consideration of local quality by sampling patches at their raw resolution and covers global quality with contextual

---

[4]https://drive.google.com/open?id=1M_4hN-beZpa-eiYCvIE7hsORjF18LEYU

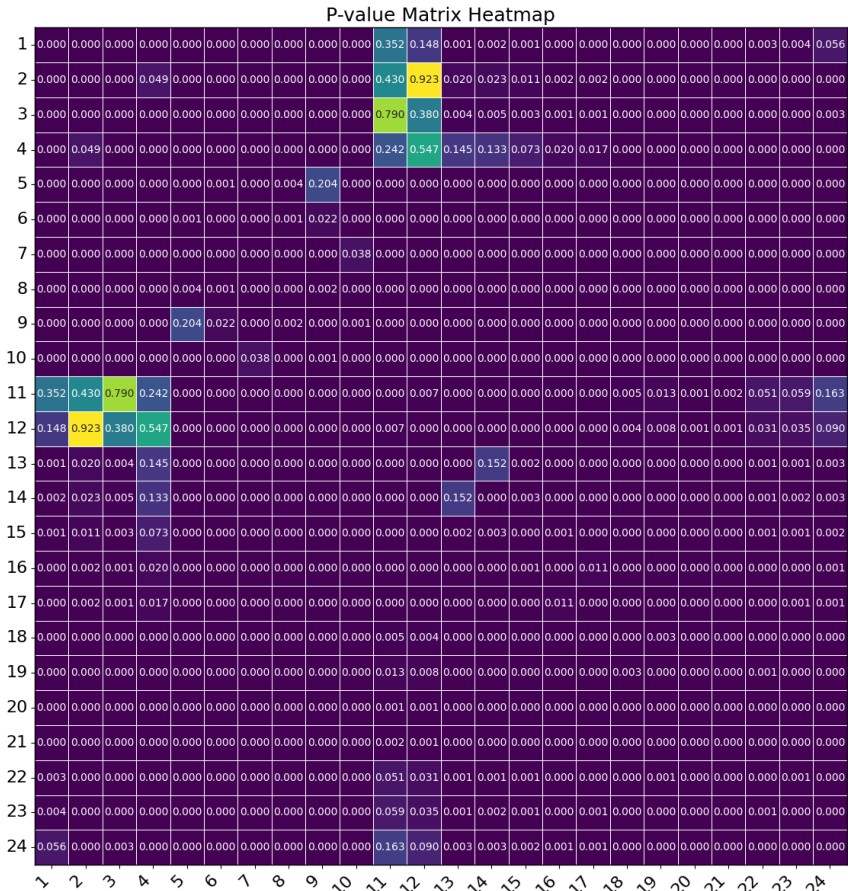

Figure 15: Statistical significance comparison among different methods on GAIA dataset. Most p-values are less than 0.001. The methods denoted by '1'-'24' are USDL, ACTION-NET, CoRe, TSA, Subject Consistency, Motion Smoothness, Dynamic Degree, Human Action, Action-Score, Flow-Score, TLVQM, VIDEVAL, VSFA, BVQA, SimpleVQA, FAST-VQA, DOVER, CLIPScore-ViT-B/16, CLIPScore-ViT-B/32, CLIPScore-ViT-B/32-LAION, CLIPScore-ViT-L/14, BLIPScore, LLaVAScore, and InternLMScore, respectively. Zoom-in for better visualization.

relations via mini-patches sampled in uniform grids. It overcomes the high computational costs when evaluating high-resolution videos. We used the officially released FAST-VQA-B model and retrained on our GAIA.

**DOVER** [113] is a disentangled objective video quality evaluator that learns the quality of videos based on technical and aesthetic perspectives. We directly used the official code without specific changes.

### C.5 Video-Text Alignment Metrics

**CLIPScore** [44] is an image captioning metric, which is widely used to evaluate T2I/T2V models. It passes both the image and the candidate caption through their respective feature extractors, then computing the cosine similarity of the resultant embeddings as the predicted score. **BLIPScore**, **LLaVAScore**, and **InternLMScore** replace CLIP with more advanced VLMs, *i.e.*, BLIP [61], LLAVA-1.5-7B [65], and Internlm-XComposer2-VL [29]. For these metrics, we uniformly sample 8 frames while rescaling them at a fixed height of 520 as input, and take the averaged frame-wise score as final results.

Table 9: 95% confidence intervals (CI) of evaluated methods. Supporting the conclusion obtained in Tab. 5 in the main paper.

| Method | $\frac{\overline{SRCC+PLCC}}{2}$ | 95% CI |
|---|---|---|
| USDL | 0.4319 | [0.4222, 0.4415] |
| ACTION-NET | 0.4668 | [0.4582, 0.4753] |
| CoRe | 0.4456 | [0.4373, 0.4538] |
| TSA | 0.4804 | [0.4619, 0.4990] |
| Subject Consistency | 0.2186 | [0.2032, 0.2340] |
| Motion Smoothness | 0.1827 | [0.1594, 0.2061] |
| Dynamic Degree | 0.0944 | [0.0758, 0.1129] |
| Human Action | 0.2713 | [0.2550, 0.2876] |
| Action-Score | 0.2429 | [0.2136, 0.2722] |
| Flow-Score | 0.1256 | [0.1054, 0.1458] |
| TLVQM | 0.4509 | [0.4135, 0.4882] |
| VIDEVAL | 0.4648 | [0.4240, 0.5055] |
| VSFA | 0.5142 | [0.4834, 0.5450] |
| BVQA | 0.5186 | [0.4835, 0.5537] |
| SimpleVQA | 0.5289 | [0.4926, 0.5651] |
| FAST-VQA | 0.5450 | [0.5127, 0.5773] |
| DOVER | 0.5534 | [0.5161, 0.5908] |
| CLIPScore$_{\text{ViT-B/16}}$ | 0.3646 | [0.3478, 0.3813] |
| CLIPScore$_{\text{ViT-B/32}}$ | 0.3735 | [0.3540, 0.3930] |
| CLIPScore$_{\text{ViT-B/32-LAION}}$ | 0.3402 | [0.3259, 0.3545] |
| CLIPScore$_{\text{ViT-L/14}}$ | 0.3466 | [0.3309, 0.3622] |
| BLIPScore | 0.3913 | [0.3654, 0.4172] |
| LLaVAScore | 0.3944 | [0.3691, 0.4197] |
| InternLMScore | 0.4124 | [0.3883, 0.4365] |

# D  Extended Results

In this section, we include more observations from the evaluations on the GAIA dataset.

**Whether CLIP-based Metrics Excel in Assessing Action Quality?** We notice that CLIPScore achieves about 0.38 SRCC and PLCC in the action completeness perspective (Tab. 5), which shows a low correlation with human perception. Although CLIP is not tuned for fine-grained actions, it may work for some coarse-grained actions, as the action itself is also related to the context of scene. We thereby conduct extra experiments to evaluate CLIPScore on three subsets of the GAIA dataset from coarse-grained actions (whole-body) to fine-grained actions (hand and facial). The results are shown in Tab. 10. We can observe that CLIPScore performs significantly worse on the facial subset (an average SRCC of 0.184, 0.194, and 0.239 in terms of subject quality, action completeness, and action-scene interaction, respectively.) than the whole-body (an average SRCC of 0.345, 0.381, and 0.378) and hand subsets. The results further demonstrate the above conjecture that CLIPScore is not appropriate for the assessment of fine-grained actions such as facial actions. Moreover, CLIPScore performs relatively better in action completeness than the subject quality perspective. As discussed in the main paper (Sec. 4.2), we conjecture that such alignment-based metrics are intrinsically sensitive to global high-level vision information (action-related semantics) rather than low-level generative flaws (*e.g.*, blur, noise, textures) that can severely affect the subject quality.

**Whether the Combination of Different Metrics can Improve the Perceptual Consistency of Action Quality?** We test several different combinations of existing metrics for comparison. As shown in Tab. 11, in most cases, the performance of the combined one is within the best performance of a single one. Surprisingly, we found a performance gain when combining different variants of CLIPScore. We hypothesize that this is due to the spatial feature compensation provided by the different convolutional kernel sizes. Moreover, we observe that combining VSFA with "Human Action" or "Flow-Score" did not yield performance improvements, rather, it resulted in a decrease in SRCC/PLCC scores. We attributed it to different scales of predicted scores, since Flow-Score is an optical flow-based metric. Adding VSFA with three variants of CLIPScore shows better SRCC/PLCC on all three perspectives compared to their single forms. As mentioned in the main paper, VQA methods perform better on subject quality than action completeness and action-scene interaction

Table 10: Performance comparison on coarse-grained actions (whole-body) and fine-grained actions (hand and facial) from GAIA dataset.

| Dimension Metrics | Subset | Subject SRCC↑ | Subject PLCC↑ | Completeness SRCC↑ | Completeness PLCC↑ | Interaction SRCC↑ | Interaction PLCC↑ |
|---|---|---|---|---|---|---|---|
| CLIPScore (ViT-B/16) | Whole-body | 0.3381 | 0.3293 | 0.3732 | 0.3656 | 0.3698 | 0.3557 |
| | Hand | 0.3167 | 0.3084 | 0.3649 | 0.3564 | 0.3361 | 0.3234 |
| | Facial | 0.2221 | 0.2326 | 0.2307 | 0.2525 | 0.2711 | 0.2861 |
| CLIPScore (ViT-B/32) | Whole-body | 0.3848 | 0.3753 | 0.4208 | 0.4128 | 0.4168 | 0.4023 |
| | Hand | 0.3835 | 0.3788 | 0.4159 | 0.4139 | 0.3964 | 0.3910 |
| | Facial | 0.1556 | 0.1596 | 0.1747 | 0.1859 | 0.2175 | 0.2201 |
| CLIPScore (ViT-L/14) | Whole-body | 0.3135 | 0.3055 | 0.3499 | 0.3411 | 0.3481 | 0.3301 |
| | Hand | 0.3392 | 0.3269 | 0.3639 | 0.3499 | 0.3373 | 0.3219 |
| | Facial | 0.1743 | 0.1806 | 0.1775 | 0.1927 | 0.2294 | 0.2359 |

Table 11: Results for the combination of different metrics on the GAIA dataset.

| Dimension Metrics | Subject SRCC↑ | Subject PLCC↑ | Completeness SRCC↑ | Completeness PLCC↑ | Interaction SRCC↑ | Interaction PLCC↑ |
|---|---|---|---|---|---|---|
| Human Action | **0.2453** | **0.2369** | **0.2895** | **0.2812** | **0.2861** | **0.2743** |
| Action-Score | 0.2023 | 0.1823 | 0.2867 | 0.2623 | 0.2689 | 0.2432 |
| Flow-Score | 0.1471 | 0.1541 | 0.0816 | 0.1273 | 0.1041 | 0.1309 |
| Human Action+Action-Score | 0.1530 | 0.1355 | 0.2333 | 0.2098 | 0.2156 | 0.1912 |
| Human Action+Flow-Score | 0.1567 | 0.1550 | 0.0940 | 0.1293 | 0.1155 | 0.1324 |
| Action-Score+Flow-Score | 0.1199 | 0.1464 | 0.0439 | 0.1175 | 0.0679 | 0.1214 |
| Human Action+Action-Score+Flow-Score | 0.1279 | 0.1484 | 0.0530 | 0.1198 | 0.0767 | 0.1237 |
| VSFA | 0.1934 | 0.1917 | 0.1379 | 0.1322 | 0.1602 | 0.1658 |
| VSFA+Human Action | 0.0836 | 0.0790 | 0.0059 | 0.0142 | 0.0135 | 0.0096 |
| VSFA+Action-Score | **0.2599** | **0.2531** | **0.3149** | **0.3046** | **0.3054** | **0.2939** |
| VSFA+Flow-Score | 0.1309 | 0.1506 | 0.0714 | 0.1253 | 0.0914 | 0.1283 |
| TSA | 0.4435 | 0.4477 | 0.4963 | 0.4981 | 0.4941 | 0.4953 |
| DOVER | **0.6173** | **0.6301** | **0.5198** | **0.5323** | **0.5164** | **0.5278** |
| TSA + DOVER | 0.5744 | 0.5831 | 0.5068 | 0.5147 | 0.5081 | 0.5158 |
| CLIPScore-B/16 | 0.3360 | 0.3314 | 0.3841 | 0.3777 | 0.3753 | 0.3632 |
| CLIPScore-B/32 | 0.3398 | 0.3330 | 0.3944 | 0.3871 | 0.3875 | 0.3821 |
| CLIPScore-L/14 | 0.3211 | 0.3156 | 0.3657 | 0.3574 | 0.3585 | 0.3426 |
| CLIPScore-B/16+CLIPScore-B/32 | 0.3746 | **0.3698** | **0.4234** | **0.4172** | **0.4148** | **0.4028** |
| CLIPScore-B/16+CLIPScore-L/14 | 0.3479 | 0.3428 | 0.3967 | 0.3893 | 0.3878 | 0.3738 |
| CLIPScore-B/32+CLIPScore-L/14 | **0.3747** | 0.3687 | 0.4218 | 0.4145 | 0.4140 | 0.3998 |
| CLIPScore-B/16+CLIPScore-B/32+CLIPScore-L/14 | 0.3734 | 0.3681 | 0.4227 | 0.4157 | 0.4140 | 0.4006 |
| VSFA+CLIPScore-B/16 | 0.3782 | 0.3733 | 0.4014 | 0.3990 | 0.3984 | 0.3906 |
| VSFA+CLIPScore-B/32 | **0.4162** | **0.4120** | **0.4377** | **0.4355** | **0.4364** | **0.4288** |
| VSFA+CLIPScore-L/14 | 0.3651 | 0.3582 | 0.3826 | 0.3793 | 0.3821 | 0.3709 |
| VSFA+CLIPScore-B/16+CLIPScore-B/32 | 0.4004 | 0.3938 | 0.4361 | 0.4303 | 0.4308 | 0.4192 |
| CLIPScore-B/16+CLIPScore-B/32+Human Action | 0.3585 | 0.3581 | 0.4041 | 0.4027 | 0.3960 | 0.3885 |

perspectives, which is opposed to CLIPScore. Therefore, combining these two kinds of metrics could effectively improve the subjective consistency of results. This observation provides intuition for the future development of better AQA methods. Additionally, CLIPScore and its variants outperform the other methods under zero-shot settings. This result suggests that considering both spatial and textual features to better associate visual features with scene descriptions is helpful in predicting action quality.

Indeed, applying a combination of multiple methods is less efficient in practical applications. In the future, we will explore the structure of different models and investigate the possibility of fusing them at the module level in an end-to-end way to better predict the action quality.

Table 12: Categories of the 400 whole-body actions in our proposed GAIA.

| Class | Action Keyword | | | |
|---|---|---|---|---|
| Arts and crafts | arranging flowers
drawing
weaving basket | blowing glass
knitting | brush painting
making jewelry | clay pottery making
spray painting |
| Athletics – jumping | high jump
pole vault | hurdling
triple jump | long jump | parkour |
| Athletics – throwing + launching | archery
javelin throw | catching or throwing frisbee
throwing axe | disc golfing
throwing ball | hammer throw
throwing discus |
| Auto maintenance | changing oil | changing wheel | checking tires | pumping gas |
| Ball sports | bowling
kicking field goal
playing basketball
shooting goal (soccer) | dodgeball
kicking soccer ball
playing kickball
shot put | dribbling basketball
passing American football (in game)
playing volleyball | dunking basketball
passing American football (not in game)
shooting basketball |
| Body motions | baby waking up
stretching leg
lunge | bending back
swinging legs | cracking neck
exercising arm | stretching arm
exercising with an exercise ball |
| Cleaning | cleaning floor
cleaning toilet
mopping floor
washing dishes | cleaning gutters
cleaning windows
setting table | cleaning pool
doing laundry
shining shoes | cleaning shoes
making bed
sweeping floor |
| Cloths | bandaging
tying knot (not on a tie) | folding clothes
tying tie | ironing | tying bow tie |
| Communication | answering questions
giving or receiving award
sign language interpreting | auctioning
laughing
testifying | celebrating
news anchoring | crying
presenting weather forecast |
| Cooking | baking cookies
cooking egg
cutting watermelon
making a cake
making tea
scrambling eggs | barbequing
cooking on campfire
flipping pancake
making a sandwich
peeling apples
tossing salad | breading or breadcrumbing
cooking sausages
frying vegetables
making pizza
peeling potatoes | cooking chicken
cutting pineapple
grinding meat
making sushi
picking fruit |
| Dancing | belly dancing
country line dancing
dancing macarena
robot dancing
tap dancing | breakdancing
dancing ballet
jumpstyle dancing
salsa dancing
zumba | capoeira
dancing charleston
krumping
swing dancing | cheerleading
dancing gangnam style
marching
tango dancing |
| Eating + drinking | bartending
drinking shots
eating chips
eating spaghetti
tasting food | dining
eating burger
eating doughnuts
eating watermelon | drinking
eating cake
eating hotdog
opening bottle | drinking beer
eating carrots
eating ice cream
tasting beer |
| Electronics | assembling computer
using remote controller (not gaming) | playing controller | texting | using computer |
| Garden + plants | blowing leaves
decorating the christmas tree
trimming trees | carving pumpkin
egg hunting
watering plants | chopping wood
mowing lawn | climbing tree
planting trees |
| Golf | golf chipping | golf driving | golf putting | |
| Gymnastics | bouncing on trampoline
vault
yoga | cartwheeling
bench pressing | gymnastics tumbling
doing aerobics | somersaulting
situp |
| Hair | braiding hair
getting a haircut | brushing hair
shaving head | curling hair
shaving legs | fixing hair
washing hair |
| Hands | air drumming
finger snapping | applauding
pumping fist | clapping
drumming fingers | cutting nails |
| Head + mouth | balloon blowing
headbanging
smoking
sticking tongue out | beatboxing
headbutting
smoking hookah
whistling | blowing nose
shaking head
sneezing
yawning | blowing out candles
singing
sniffing
gargling |
| Heights | abseiling
paragliding
swinging on something | bungee jumping
rock climbing
trapezing | climbing a rope
skydiving | climbing ladder
slacklining |
| Interacting with animals | bee keeping
feeding goats
milking cow
riding elephant
training dog | catching fish
grooming dog
petting animal (not cat)
riding mule
walking the dog | feeding birds
grooming horse
petting cat
riding or walking with horse | feeding fish
holding snake
riding camel
shearing sheep |
| Juggling | contact juggling
juggling soccer ball | hula hooping
spinning poi | juggling balls | juggling fire |
| Makeup | applying cream
getting a tattoo | doing nails | dying hair | filling eyebrows |
| Martial arts | arm wrestling
punching person
wrestling | drop kicking
side kick | high kick
sword fighting | punching bag
tai chi |
| Miscellaneous | digging
moving furniture
unloading truck | extinguishing fire
spraying | garbage collecting
stomping grapes | laying bricks
tapping pen |
| Mobility – land | crawling baby
hoverboarding
pushing cart
riding scooter
skateboarding | driving car
jogging
pushing wheelchair
riding unicycle
surfing crowd | driving tractor
motorcycling
riding a bike
roller skating
using segway | faceplanting
pushing car
riding mountain bike
running on treadmill
waiting in line |
| Mobility – water | crossing river
snorkeling | diving cliff
springboard diving | jumping into pool
water sliding | scuba diving |
| Music | busking
playing cello
playing drums
playing harp
playing recorder
playing ukulele
strumming guitar | playing accordion
playing clarinet
playing flute
playing keyboard
playing saxophone
playing violin
tapping guitar | playing bagpipes
playing cymbals
playing guitar
playing organ
playing trombone
playing xylophone | playing bass guitar
playing didgeridoo
playing harmonica
playing piano
playing trumpet
recording music |
| Paper | bookbinding
opening present
shredding paper | counting money
reading book
unboxing | folding napkins
reading newspaper
wrapping present | folding paper
ripping paper
writing |
| Personal hygiene | brushing teeth
washing hands | taking a shower | trimming or shaving beard | washing feet |

Table 13: Extension of Tab. 12.

| Class | Action Keyword | | | |
|---|---|---|---|---|
| Playing games | flying kite
playing paintball
skipping rope | hopscotch
playing poker
tossing coin | playing cards
riding mechanical bull
playing monopoly | playing chess
rock scissors paper
shuffling cards |
| Racquet + bat sports | catching or throwing baseball
playing badminton | catching or throwing softball
playing cricket | hitting baseball
playing squash or racquetball | hurling (sport)
playing tennis |
| Snow + ice | biking through snow
ice fishing
shoveling snow
skiing slalom
snowmobiling | bobsledding
ice skating
ski jumping
sled dog racing
tobogganing | hockey stop
making snowman
skiing (not slalom or crosscountry)
snowboarding | ice climbing
playing ice hockey
skiing crosscountry
snowkiting |
| Swimming | swimming backstroke | swimming breast stroke | swimming butterfly stroke | |
| Touching person | carrying baby
massaging feet
slapping | hugging
massaging legs
tickling | kissing
massaging person's head | massaging back
shaking hands |
| Using tools | bending metal
plastering
welding | blasting sand
sanding floor | building cabinet
sharpening knives | building shed
sharpening pencil |
| Water sports | canoeing or kayaking
sailing | jetskiing
surfing water | kitesurfing
water skiing | parasailing
windsurfing |
| Waxing | waxing back | waxing chest | waxing eyebrows | waxing legs |
| Weightlifting | pull ups
front raises | push up
snatch weight lifting | clean and jerk
squat | deadlifting |

Table 14: Categories of the 83 hand actions in our proposed GAIA.

| Class | Action Keyword | | | |
|---|---|---|---|---|
| Move | Wave palm towards right
Wave palm forward
Move fist upward
Move palm backward
Move palm towards left
Move fingers toward left | Wave palm towards left
Wave palm backward
Move fist downward
Move palm forward
Move palm towards right
Move fingers toward right | Wave palm downward
Wave finger towards left
Move fist towards left
Move palm upward
Move fingers upward
Move fingers forward | Wave palm upward
Wave finger towards right
Move fist towards right
Move palm downward
Move fingers downward |
| Zoom | Zoom in with two fists | Zoom out with two fists | Zoom in with two fingers | Zoom out with two fingers |
| Rotate | Rotate fists clockwise | Rotate fists counter-clockwise | Rotate fingers clockwise | Rotate fingers counter-clockwise |
| Open/close | Turn over palm
Put two fingers together | Rotate with palm
Take two fingers apart | Palm to fist | Fist to Palm |
| Number | Number 0
Number 4
Number 8 | Number 1
Number 5
Number 9 | Number 2
Number 6
Another number 3 | Number 3
Number 7 |
| Direction | Thumb upward
Thumbs backward | Thumb downward
Thumbs forward | Thumb towards right | Thumb towards left |
| Others | Cross index fingers
OK
Dual hands heart | Sweep cross
Pause
Bent two fingers | Sweep checkmark
Shape C
Bent three fingers | Static fist
Hold fist in the other hand
Dual fingers heart |
| Mimetic | Click with index finger
take a picture
Knock
Grab (bend all five fingers)
Applaud | Sweep diagonal
Make a phone call
Beckon
Walk | Measure (distance)
Wave hand
Trigger with thumb
Gather fingers | Sweep circle
Wave finger
Trigger with index finger
Snap fingers |
| Surprised | curiosity
surprise | desire | approval | realization |
| Fearful | confusion
caring | fear | nervousness | relief |
| Disgusted | disgust | embarrassment | | |
| Happy | amusement
optimism | love
pride | joy
admiration | excitement
gratitude |
| Sad | disappointment
sadness | disapproval | grief | remorse |
| Angry | anger | annoyance | | |

Table 15: Categories of the 27 facial actions in our proposed GAIA.

| Class | Action Keyword | | | | |
|---|---|---|---|---|---|
| Surprised | curiosity | desire | approval | realization | surprise |
| Fearful | confusion | fear | nervousness | relief | caring |
| Disgusted | disgust | embarrassment | | | |
| Happy | amusement
pride | love
admiration | joy
gratitude | excitement | optimism |
| Sad | disappointment | disapproval | grief | remorse | sadness |
| Angry | anger | annoyance | | | |

