# Supplementary Materials for GAIA: Rethinking Action Quality Assessment for AI-Generated Videos

**Zijian Chen**[1], **Wei Sun**[1*], **Yuan Tian**[1], **Jun Jia**[1], **Zicheng Zhang**[1],
**Jiarui Wang**[1], **Ru Huang**[2], **Xiongkuo Min**[1*], **Guangtao Zhai**[1*], **Wenjun Zhang**[1]
[1]Shanghai Jiao Tong University  [2]East China University of Science and Technology
[*]Corresponding authors

## 1 Dataset Documentation

Here we provide documentation for our dataset in the common datasheet format [11].

### 1.1 Motivation

**For what purpose was the dataset created? Was there a specific task in mind? Was there a specific gap that needed to be filled?**

We produced the dataset to fill the gaps of current research on action quality assessment (AQA) in AI-generated videos (AIGV) and evaluate the existing automatic action-quality and video-quality metrics. To the best of our knowledge, it is the first AQA dataset that evaluates action quality from causal reasoning-based perspectives. Investigations of the action quality assessment commonly conduct for specific-domain in real-world scenarios (*e.g.*, sport events and health care). Our goal is to create an AQA dataset that focuses on AI-generated scenarios.

**Who created the dataset (for example, which team, research group) and on behalf of which entity (for example, company, institution, or organization)?**

The dataset was a joint effort by Zijian Chen[1], Wei Sun[1], Yuan Tian[1], Jun Jia[1], Zicheng Zhang[1], Jiarui Wang[1], Ru Huang[2], Xiongkuo Min[1], Guangtao Zhai[1], and Wenjun Zhang[1]. The authors[1] are researchers affiliated with the Institute of Image Communication and Information Processing, Shanghai Jiao Tong University. The author[2] is with the School of Information Science & Engineering, East China University of Science and Technology.

**Who funded the creation of the dataset? If there is an associated grant, please provide the name of the grantor and the grant name and number.**

This work was supported in part by the China Postdoctoral Science Foundation under Grant Number 2023TQ0212 and 2023M742298, in part by the Postdoctoral Fellowship Program of CPSF under Grant Number GZC20231618, in part by the Shanghai Pujiang Program under Grant 22PJ1407400, and in part by the National Natural Science Foundation of China under Grant 62271312, 62301316, 62101325 and 62101326.

### 1.2 Composition

**What do the instances that comprise the dataset represent (for example, documents, photos, people, countries)? Are there multiple types of instances (for example, movies, users, and ratings; people and interactions between them; nodes and edges)?**

The instances represent video files with corresponding subjective action quality scores. Each video was generated by a text-to-video model using a prompt that contained the action keyword.

**How many instances are there in total (of each type, if appropriate)?**

The dataset has a total of 9,180 AI-generated videos with 971,244 reliable ratings (an average of 105.8 ratings per video (35.27 per dimension)).

**Does the dataset contain all possible instances or is it a sample (not necessarily random) of instances from a larger set? If the dataset is a sample, then what is the larger set? Is the sample representative of the larger set (for example, geographic coverage)?**

The dataset contains all generated instances.

**What data does each instance consist of? "Raw" data (for example, unprocessed text or images) or features?**

Each instance is a video file in .mp4 format. Each subjective score is a floating-point number. Each action keyword and the corresponding prompt are in text format.

**Is there a label or target associated with each instance?**

Each video has three subjective scores that represent the evaluated three action quality-related perspectives (*i.e.*, *subject quality*, *action completeness*, and *action-scene interaction*) with the corresponding prompt and action keyword.

**Is any information missing from individual instances?**

No.

**Are relationships between individual instances made explicit (for example, users' movie ratings, social network links)?**

We divided the action videos into groups on the basis of their action categories and their respective text-to-video models.

**Are there recommended data splits (for example, training, development/validation, testing)?**

No. In the process of benchmarking the current action quality assessment metrics, we randomly split the training-testing set at a ratio of 8:2 ten times.

**Are there any errors, sources of noise, or redundancies in the dataset?**

No

**Is the dataset self-contained, or does it link to or otherwise rely on external resources (for example, websites, tweets, other datasets)?**

The dataset is self-contained.

**Does the dataset contain data that might be considered confidential (for example, data that is protected by legal privilege or by doctor-patient confidentiality, data that includes the content of individuals' non-public communications)?**

No. All videos in our GAIA dataset are generated by text-to-video models according to the given action prompt.

**Does the dataset contain data that, if viewed directly, might be offensive, insulting, threatening, or might otherwise cause anxiety?**

No. We have organized experts to manually check the compliance and legality of the adopted action prompts as well as the generated videos to avoid such issues.

## 1.3 Collection Process

**How was the data associated with each instance acquired? Was the data directly observable (for example, raw text, movie ratings), reported by subjects (for example, survey responses), or indirectly inferred/derived from other data (for example, part-of-speech tags, model-based guesses for age or language)? If the data was reported by subjects or indirectly inferred/derived from other data, was the data validated/verified?**

We formed the dataset using 18 popular text-to-video models including 11 lab-studies and 7 commercial applications, listed in Tab 1.

Table 1: URLs for the adopted text-to-video models.

| Methods | URL |
|---|---|
| Text2Video-Zero [14] | https://github.com/Picsart-AI-Research/Text2Video-Zero |
| ModelScope [17] | https://modelscope.cn/models/iic/text-to-video-synthesis/summary |
| ZeroScope [8] | https://huggingface.co/cerspense/zeroscope_v2_576w |
| LaVie [18] | https://github.com/Vchitect/LaVie |
| Show-1 [20] | https://github.com/showlab/Show-1 |
| Hotshot-XL [15] | https://github.com/hotshotco/Hotshot-XL |
| AnimateDiff [12] | https://github.com/guoyww/AnimateDiff |
| VideoCrafter1-512 [9] | https://github.com/AILab-CVC/VideoCrafter |
| VideoCrafter1-1024 [9] | https://github.com/AILab-CVC/VideoCrafter |
| VideoCrafter2 [10] | https://github.com/AILab-CVC/VideoCrafter |
| Mora [19] | https://github.com/lichao-sun/Mora |
| Gen-2 [1] | https://research.runwayml.com/gen2 |
| Genmo [2] | https://www.genmo.ai |
| Pika [6] | https://pika.art/home |
| NeverEnds [5] | https://neverends.life |
| MoonValley [3] | https://moonvalley.ai |
| Morph Studio [4] | https://www.morphstudio.com |
| Stable Video [7] | https://www.stablevideo.com/welcome |

Content types: 400 whole-body actions, 83 hand actions, and 27 facial actions. Detailed category of each action keyword in our GAIA dataset are listed in Tab. 2, Tab. 3, Tab. 4, and Tab. 5

**What mechanisms or procedures were used to collect the data (for example, hardware apparatuses or sensors, manual human curation, software programs, software APIs)? How were these mechanisms or procedures validated?**

The subjective experiments (labeling) involved a training/pre-labeling and a main study processes. During the training, we instructed all participants to have a clear and consistent understanding of all evaluated aspects and tested their eligibility via a 30-video pre-labeling. In the tutorial for each dimension, participants are guided to rate 10 generated-real video pairs with the same caption. Their answer is compared with ground-truth ratings that were developed by multiple experts. Raters needed to achieve at least 75% ratings that satisfied $|ground\_truth - rating| < 1.5\sigma_{expert}$ to move on to the formal study. We adopted a single-stimulus methodology in this evaluation and asked participants to focus on the given action keyword as well as the corresponding prompt and evaluate three action-related dimensions of AI-generated videos, *i.e.*, *subject quality*, *action completeness*, and *action-scene interaction*, by dragging the slide button at a $[0, 100]$ continuous rating scale (see Fig. 1). In addition to the above pre-labeling and in-process check trial, we also performed line clickers examination and calculated the inter-annotator agreement metric (Krippendorff's $\alpha$ [13]) as well as the SRCC score using bootstrapping to ensure the annotation quality.

**If the dataset is a sample from a larger set, what was the sampling strategy (for example, deterministic, probabilistic with specific sampling probabilities)?**

Not applicable.

**Who was involved in the data collection process (for example, students, crowdworkers, contractors) and how were they compensated (for example, how much were crowdworkers paid)?**

We recruit a total of 54 participants to participate in our subjective experiments. Each participant was compensated $12 for each session (rating 300 videos in three dimensions) according to the current ethical standard [16].

**Over what timeframe was the data collected? Does this timeframe match the creation timeframe of the data associated with the instances (for example, recent crawl of old news articles)?**

The collection of source content in GAIA dataset took place nearly one years (from March 2023 to March 2024). The timeframe is align with the released date of the selected text-to-video models. The subjective experiments took over a month (from March 2024 to April 2024) to complete.

**Were any ethical review processes conducted (for example, by an institutional review board)?**

We have organized experts to manually check the compliance and legality of the adopted action prompts as well as the generated videos to avoid ethical issues.

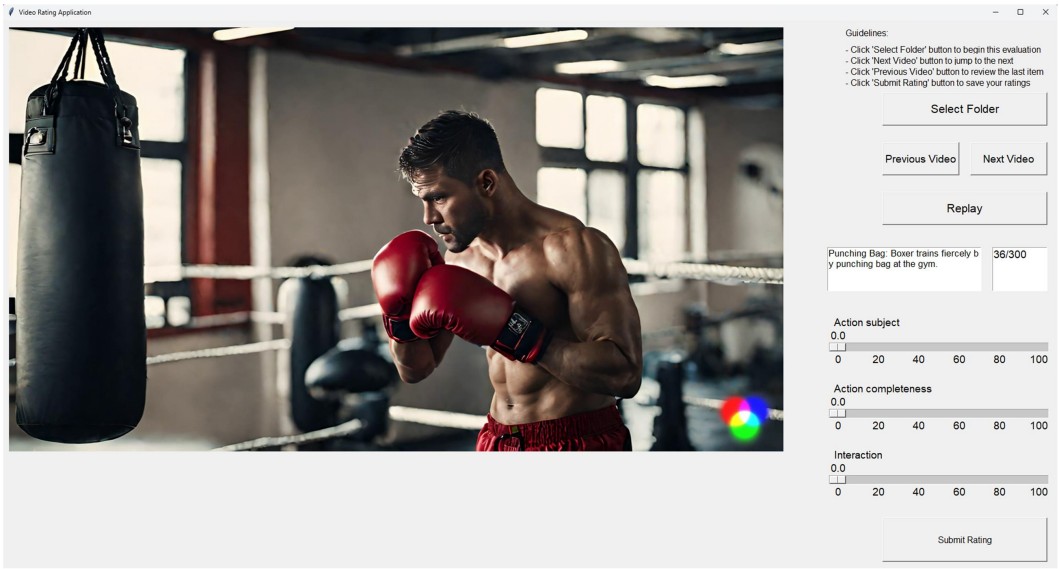

Figure 1: **Screenshot of the rating interface for human evaluation.** Participants are instructed to rate three action-related dimensions of AI-generated videos, *i.e.*, *subject quality*, *action completeness*, and *action-scene interaction*, based on the given action keyword and prompt.

## 1.4 Preprocessing/Cleaning/Labeling

**Was any preprocessing/cleaning/labeling of the data done (for example, discretization or bucketing, tokenization, part-of-speech tagging, SIFT feature extraction, removal of instances, processing of missing values)?**

We directly use the generated videos from text-to-video models as source content without any change. For the collected raw user-rated scores, we performed Z-score normalization to avoid inter-annotator scoring biases.

**Was the "raw" data saved in addition to the preprocessed/cleaned/labeled data (for example, to support unanticipated future uses)?**

Yes, we saved the raw user-rated scores for all dimensions.

**Is the software that was used to preprocess/clean/label the data available?**

We conducted the subjective experiments in-lab. All data was labeled through a self-designed program based on Python Tkinter (Fig. 1).

## 1.5 Uses

**Has the dataset been used for any tasks already?**

The dataset has served in two separate tasks:

- Measurement of action-quality metrics for AI-generated videos along with calculation of the correlation between the predicted scores and subjective scores.

- Comparing the performance of different video generation models in generating rational, artifact-free, and temporally consistent actions.

**Is there a repository that links to any or all papers or systems that use the dataset?**

The benchmark for action quality assessment in AI-generated videos is available through `https://github.com/zijianchen98/GAIA`.

**What (other) tasks could the dataset be used for?**

The dataset can be used to train or validate a new action quality assessment metric and determine whether it can reliably meet most action scenarios.

**Is there anything about the composition of the dataset or the way it was collected and preprocessed/cleaned/labeled that might impact future uses? For example, is there anything that a dataset consumer might need to know to avoid uses that could result in unfair treatment of individuals or groups (for example, stereotyping, quality of service issues) or other risks or harms (for example, legal risks, financial harms)? Is there anything a dataset consumer could do to mitigate these risks or harms?**

We applied a novel causal reasoning-based subjective evaluation strategy to collect opinions for assessing action quality. Consumers are recommended not to use the subjective score of a single dimension to compare the action quality since it is not fair.

**Are there tasks for which the dataset should not be used?**

No.

## 1.6 Distribution

**Will the dataset be distributed to third parties outside of the entity (for example, company, institution, organization) on behalf of which the dataset was created?**

The GAIA dataset is available to everyone.

**How will the dataset be distributed (for example, tarball on website, API, GitHub)? Does the dataset have a digital object identifier (DOI)?**

The GAIA dataset is accessible through `https://github.com/zijianchen98/GAIA`.

**When will the dataset be distributed?**

We are now preparing to release the dataset. All information will be noted at a Github repository `https://github.com/zijianchen98/GAIA`.

**Will the dataset be distributed under a copyright or other intellectual property (IP) license, and/or under applicable terms of use (ToU)?**

The **GAIA** dataset is released under the **CC BY 4.0** license, which includes all associated AIGVs, scores, and their corresponding action prompts.

**Have any third parties imposed IP-based or other restrictions on the data associated with the instances?**

No.

**Do any export controls or other regulatory restrictions apply to the dataset or to individual instances?**

No.

## 1.7 Maintenance

**Who will be supporting/hosting/maintaining the dataset?**

The Shanghai Jiao Tong University MultiMedia Lab hosts the dataset. The authors of this paper support the proposed GAIA dataset and the benchmark evaluated in this paper.

**How can the owner/curator/manager of the dataset be contacted (for example, email address)?**

Contact the first author of this paper Mr. Zijian Chen by `zijian.chen@sjtu.edu.cn`, or the corresponding author Prof. Guangtao Zhai by `zhaiguangtao@sjtu.edu.cn`.

**Is there an erratum?**

No.

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

**Will the dataset be updated (for example, to correct labeling errors, add new instances, delete instances)? If so, please describe how often, by whom, and how updates will be communicated to dataset consumers (for example, mailing list, GitHub)?**

We plan to extend the dataset as the generative model evolves to ensure benchmark results with the highest statistical credibility and the latest observations. Such updates will be rare, as they involve subjective evaluation, a time-consuming task that requires extensive preparation. This does not affect the user's use of the current dataset. All the new information will be on the dataset website https://github.com/zijianchen98/GAIA.

**If the dataset relates to people, are there applicable limits on the retention of the data associated with the instances (for example, were the individuals in question told that their data would be retained for a fixed period of time and then deleted)?**

Not applicable.

**Will older versions of the dataset continue to be supported/hosted/maintained?**

We do not intend to create a version history.

**If others want to extend/augment/build on/contribute to the dataset, is there a mechanism for them to do so? Will these contributions be validated/verified? If not, why not? Is there a process for communicating/distributing these contributions to dataset consumers?**

We encourage future researchers to share their ideas on extending our dataset to cover more action cases and model variants while providing more reliable results. We recommend people contacting us by zijian.chen@sjtu.edu.cn to collaborate on the subjective quality evaluation. It is recommended to build a more large-scale and comprehensive dataset based on our dataset.