# OpenReview forum: "GAIA: Rethinking Action Quality Assessment for AI-Generated Videos"
_NeurIPS.cc/2024/Datasets_and_Benchmarks_Track — NeurIPS 2024 Track Datasets and Benchmarks Spotlight_

### Official Review · Reviewer_jQF6 · 2024-07-23
**Opens Up a New Application for Action Quality Assessment**

**Rating:** 8
**Confidence:** 5
**Correctness:** Correct in my assessment.
**Clarity:** Writing is okay, but can definitely b…

**Review:**

Currently, Action Quality Assessment (AQA) is employed to assess or quantify skills or how well actions are executed. This paper introduces a novel use of AQA---use it to measure the quality of generated videos. In this day and age of generative AI, I think this is an important contribution and can open up new line of work for AQA. This works introduces a pilot dataset and study in this direction. Dataset contains 9180 samples with about 100 ratings per sample. 18 video generators were evaluated.

**Strengths:**

- opens up a new application of AQA
- good sized dataset with diverse actions; and large number of ratings
- detailed study

**Additional Feedback:**

- L167-170: Why the ratings were removed and how does removing exactly help? It is not clear to me, the reasoning provided did not make it clear to me.
- L184-185: Not sure why/how Z-score normalization helps?
- L311: Can you please explain why CLIPScore works well for action completeness? CLIP is not tuned for fine-grained actions/attributes...assessing action completeness in my understanding requires more finegrained understanding of the action. So, not sure why/how CLIPScore works well for this?

**Documentation:**

Yes.

**Ethics:**

NA.

**Limitations:**

NA.

**Opportunities For Improvement:**

- Writing can use some work. For example, the opening paragraph can be written with focus, and connections between the parts could be improved. Typo: L132. Ending sentence in L266 does not add value---could be removed or its connection should be improved.
- Differentiating this work with traditional image/video quality metrics is important, and unclear in the current version of the paper (L90-96). It's a good opportunity to further show the importance of this work.

**Relation To Prior Work:**

Can be improved as mentioned in Improvements section.

**Summary And Contributions:**

Currently, Action Quality Assessment (AQA) is employed to assess or quantify skills or how well actions are executed. This paper introduces a novel use of AQA---use it to measure the quality of generated videos. In this day and age of generative AI, I think this is an important contribution and can open up new line of work for AQA. This works introduces a pilot dataset and study in this direction.

---

> ### Author Rebuttal · Authors · 2024-08-15
>
> We thank the reviewer for the positive rating of our paper! We also appreciate the reviewer for acknowledging the novelty of this work and all the constructive suggestions. We hope the following clarifications can address the reviewer's concerns.
>
> **Improving writing.** Thanks for your careful reading and suggestions. We revised the typo in Line 132 and removed the ending sentence in L266. Other typos mentioned by **Reviewer 5JDo** are also revised as suggested. We have corrected these mistakes and will properly update the paper.
>
> **Differentiating this work from traditional image/video quality metrics.** We agree that the distinction between our work and traditional image/video quality metrics needs to be clarified further. In the revised manuscript, we add the following contents after line 96 to follow up on the shortcomings of the current research summarized above and explicitly differentiate our approach from conventional metrics:
>
> - Our work differs from current research in three key aspects: 1)  We created 510 distinct action prompts covering both coarse-grained and fine-grained actions, each applied with 18 T2V models for extensive assessment.  2) Our casual reasoning-based and multi-dimensional action quality evaluation offers valuable and comprehensive insights into video generation. 3) We have quantitatively validated a large amount of existing metrics that none of them performs well on the AI-generated action quality assessment task.
>
> **Why the ratings were removed and how does removing exactly help?** Line clickers are participants with an unusually high frequency for any single answer choice [1]. In the scoring quality review phase, we found 5 line clickers with over 40% of the same ratings. In other words, over 3,672 videos are rated with same scores on one or more rating dimensions (subject quality, action completeness, or action-scene interaction). In our subjective experiment, we adopt a [0, 100] continuous rating scale. Normally, the human ratings of different videos should not be the same. Such unreliable ratings, resulting from the inattentiveness of participants, deliberate action, or some other reasons, should be removed to avoid affecting the overall labeling quality and the accuracy of the final MOS scores.
>
> **how Z-score normalization helps?** Z-score nornalization is a commonly used method to avoid inter-subject scoring differences (individual biases). We normalized the raw scores of subjects to Z-scores ranging between 0 and 100 and calculated the mean of Z-scores to obtain the MOSs, which are formulated as follows:
>
> \begin{equation}
> z_{ij} = \frac{{r_{ij} - \mu _i}}{{\sigma _i}},
> \end{equation}
>
> \begin{equation}
> MO{S_j} = \frac{1}{N}\sum\limits_{i = 1}^N {\text{Res}}\left( {{z_{ij}}} \right) ,
> \end{equation}
>
> where $r_{ij}$ is the raw score given by the $i$-th participant to the $j$-th action video. $\mu _i$ and $\sigma _i$ denote the mean score and standard deviation of the $i$-th participant. $N$ is the total number of participants. $Res(\cdot)$ is the rescaling function. This practice has been widely used in building image quality assessment (IQA) or video quality assessment (VQA) datasets [2, 3]. We apologize that the current version did not sufficiently note this aspect, and we will add related references and equations in the final revision.
>
> **Performance of CLIPScore in action completeness perspective.** First, CLIPScore achieves about 0.38 SRCC and PLCC in the action completeness perspective (Tab. 5), which shows a low correlation with human perception (In most cases, we observe good correlations (SRCC, PLCC > 0.8)). We agree with the reviewer that CLIP is not tuned for fine-grained actions. However, it may work for some coarse-grained actions, as the action itself is also related to the context of scene. We conduct extra experiments to evaluate CLIPScore on three subsets of the GAIA dataset from coarse-grained actions (whole-body) to fine-grained actions (hand and facial).
> The results are shown in Tab. 1 in the uploaded PDF file. We can observe that CLIPScore performs significantly worse on the facial subset (an average SRCC of 0.184, 0.194, and 0.239 in terms of subject quality, action completeness, and action-scene interaction, respectively.) than the whole-body (an average SRCC of 0.345, 0.381, and 0.378) and hand subsets. The results further demonstrate the above conjecture that CLIPScore is not appropriate for the assessment of fine-grained actions such as facial actions.
> Moreover, CLIPScore performs relatively better in action completeness than the subject quality perspective. As discussed in Section 4.2, we conjecture that such alignment-based metrics are intrinsically sensitive to global high-level vision information (action-related semantics) rather than low-level generative flaws (e.g., blur, noise, textures) that can severely affect the subject quality.
>
> [1] Hosu V, Lin H, Sziranyi T, et al. KonIQ-10k: An ecologically valid database for deep learning of blind image quality assessment[J]. IEEE Transactions on Image Processing, 2020, 29: 4041-4056.
>
> [2] Chandler D M. Seven challenges in image quality assessment: past, present, and future research[J]. International Scholarly Research Notices, 2013, 2013(1): 905685.
>
> [3] Seshadrinathan K, Soundararajan R, Bovik A C, et al. Study of subjective and objective quality assessment of video[J]. IEEE Transactions on Image Processing, 2010, 19(6): 1427-1441.

---

> > ### Comment · Reviewer_jQF6 · 2024-08-20
> >
> > Thanks to the authors for answering my questions. I am voting to accept this paper.

---

> > > ### Author Response · Authors · 2024-08-20
> > > **Thank the reviewer**
> > >
> > > We would like to thank the reviewer for all the valuable comments and questions. We are grateful for your engagement in the rebuttal process.

---

### Official Review · Reviewer_T8oF · 2024-07-23
**Generic action quality dataset**

**Rating:** 9
**Confidence:** 5
**Correctness:** The results are convincing and claims…
**Clarity:** Yes, the paper is well written.

**Review:**

The paper describes a novel relatively large and diverse dataset with human actions that may have an impact on the development of AQA algorithms and action generation approaches. Pros: *Creation of large dataset with scores obtained with humans subjects,*Large action diversity, * Interesting comparative evaluation of methods. Cons: *Lack of discussion on the impact of prompts on the video quality, * The completeness and interaction scores seem interchangeable.

**Strengths:**

1.	The greatest contribution of this study is the large number of generated action videos used in the study and the organization of the subjective tests.
2.	A relatively large pool of compared representative methods.

**Additional Feedback:**

It is written that the benchmark “can make models pre-trained on this dataset less biased in assessing incomplete actions and irrational actions that easily appear in AI-generated scenarios.” Please clarify taking into account the results presented in this paper. Which support this claim?

**Documentation:**

The details on data collection and organization are sufficient.

**Ethics:**

There are no ethical concerns related to this study.

**Limitations:**

As the dataset consists of AI-generated videos based on text inputs (prompts), their impact on the results requires investigation.

**Opportunities For Improvement:**

1. The paper lacks a typical figure that shows MOS plotted against standard deviation (MOS), along with a discussion of the resulting shape. Figure 3 could be improved by including such figures for both the models and the entire dataset.
2. It appears that both completeness and interaction scores are quite similar. This similarity should be investigated further. Additionally, Table 5 suggests that the performance of many metrics is close for both scores. Are both completeness and interaction scores truly necessary for the analysis?
3. The text should clarify how the use of Action syllogism helped "to collect a more explainable and nuanced understanding of public perception on action assessment."
4. Since there are no existing AI-generated benchmarks that consider action quality, the method used to generate them should also be considered. In real-world datasets, the concept of "perfect" human action quality doesn't exist (as seen in Table 1 with other datasets). Consequently, some generated actions might be inherently better, or compared models could show advantages based on different prompts used for generation. The provided comparison of models is fair given the prompts used, but it's inaccurate to claim a model performs poorly simply because the prompt may not have suited its capabilities. For example, some models might require specific input phrases related to quality or a different prompt structure.
5. Please provide interpretations for the correlation values. Simply stating that the correlation is poor may be confusing for readers. In most cases, we observe weak correlations (0.3-0.5) or very weak correlations (0-0.3).

**Relation To Prior Work:**

Yes, the limitations of previous studies are discussed.

**Summary And Contributions:**

In the paper, a generic action dataset composed of AI- generated content is introduced along with three categories of subjective scores.

---

> ### Author Rebuttal · Authors · 2024-08-15
>
> We thank the reviewer for the positive rating of our paper! We also appreciate the reviewer for acknowledging the novelty of this work and all the constructive suggestions. We hope the following clarifications can address the reviewer's concerns.
>
> **Figure for standard deviation.** We added the scatter plots about MOS against standard deviation (STD), along with the five-parameter polynomial fitting plot (orange line), shown in Fig.1 of the uploaded PDF file. ***First***, there is a relatively linear distribution of STD for all three perspectives with MOS<15, suggesting that humans are more consistent in perceiving poor-quality actions. Similar observations can be found in high MOS scenarios (MOS>90). ***Second***, the trend lines reveal a peak in STD distribution when MOS$\in [20, 40]$, with a steeper decline and increase in the high MOS range (MOS>80) and low MOS range (MOS<30), respectively. We speculate that AI-generated high-quality actions are mostly consistent with people's common sense, whereas medium- and low-quality actions exhibit greater diversity, leading to a more pronounced divergence among individuals. Another plausible explanation is that this is due to the uneven distribution of high and low quality action videos in GAIA. ***Third***, the STD distribution is narrower for the subject quality dimension than for action completeness and action-scene interaction dimensions, indicating that the perception of spatial quality distortion in action is less divergent than the temporal consistency and rationality distortion.
>
> **The necessity of analyzing completeness and interaction aspects.** In this study, we decompose an action process into three parts: 1) action subject as major premise, 2) action completeness as minor premise, and 3) interaction between action and scenes as conclusion, to form the action syllogism. Each of them stands for different perspectives:
>
> - Analyzing action completeness ensures that the generated action is not only temporally coherent but also logically and narratively complete.
> - Action-scene interaction is also crucial, since it considers the spatial relationships, environmental factors, and interactions with other elements within the scene that can influence the perception of the action's quality and realism.
> - The similarity between completeness and interaction aspects lies in that they are both long-term perceptions mediated by visual memory and are easy to be affected by later video frames [1].
>
> **Clarify how the use of action syllogism helped to collect a more explainable public perception.** The action syllogism provides a logical framework that helps collect the reasoning opinions of action quality behind human perceptions, which can offer several benefits:
>
> - ***First***, syllogism is a heuristic theories that can capture opinions that could underlie intuitive responses. By breaking down an action into its constituent parts, researchers can more clearly identify and analyze the specific elements that contribute to the perceived quality of the action. [2]
> - ***Second***, such causal reasoning-based strategy is inherently align to human perception and can help in understanding how different parts of action are perceived by the public, which can lead to insights into what makes AI-generated action convincing or unconvincing.
> - ***Third***, this scheme allows for a comparative analysis of AI-generated action against natural human action, highlighting aspects where AI excels and where it may need improvement.
>
> **The impact of prompts on the generated action quality.** Considering the average length of current action prompts is 8.25 words, we intend to further explore the impact of prompt structure on the quality of the generated actions. Here, we provide three variants: 1) *Add word length to about 50 words.* 2) *Add detailed description on the action itself.* 3) *Add specialized quality suffix*. Considering the time and labor costs of conducting large-scale subjective study, we leave this part to future work.
>
> **Adding interpretations for the correlation values.** We added numeric specifications for two ambiguous statements in the current version about “poor correlation”, i.e., Line 15 in the Abstract and Line 60:  ……perform poorly with an average SRCC of 0.454, 0.191, and 0.519, respectively……
>
> **Explanation about the performance gains when pre-training on the GAIA dataset.** As mentioned in the implementation details in Appendix C, we retrained all AQA methods and VQA methods on the GAIA dataset. The action-related metrics and video-text alignment metrics were evaluated under zero-shot setting. As shown in the Tab. 5 of the main paper, AQA methods achieve an average SRCC of 0.4367, 0.4722, and 0.4664 in terms of subject quality, completeness, and interaction perspectives, respectively, while VQA methods achieve an average SRCC of 0.5811, 0.4795, and 0.4690. As a contrast, those action-related metrics with zero-shot settings only achieve an average SRCC of 0.2014, 0.1845, and 0.1918, and video-text alignment metrics achieve an average SRCC of 0.3395, 0.3954, and 0.3878. We find that models pre-trained on the GAIA dataset significantly perform better than those methods with zero-shot setting. Second, we conducted extra experiments and selected four VQA methods under zero-shot setting for comparison. As shown in the Tab.1 of the uploaded PDF file, the models with zero-shot settings significantly underperformed.
>
> [1] Rossetto L, Bailer W, Bernstein A. Considering human perception and memory in interactive multimedia retrieval evaluations[C]//International Conference on Multimedia Modeling. Cham: Springer International Publishing, 2021: 605-616.
>
> [2] Khemlani S, Johnson-Laird P N. Theories of the syllogism: A meta-analysis[J]. Psychological bulletin, 2012, 138(3): 427.

---

### Official Review · Reviewer_5JDo · 2024-07-24
**Review of GAIA**

**Rating:** 8
**Confidence:** 3

**Review:**

Overall, the methods and comparisons are exceptionally thorough. The work provides important and useful insights into AIGV models. There are a few potential areas for improvement. However, my most pressing concern is the absence of p-values or confidence intervals that support the authors’ claims and comparisons.

**Strengths:**

•	The challenge of action quality assessment is well-motivated, and this work centers itself nicely within the field of AIGV.

•	This work is timely given the number of new models released since just the beginning of 2023.

•	Thorough inclusion and discussion of many current SOTA methods and benchmarks.

•	Decomposition of human action perception is a clever and interesting idea.

•	The data appears to have been thoroughly assessed for quality.

•	The results in Table 5 are quite compelling and demonstrate the need for better metrics.

**Additional Feedback:**

N/A

**Clarity:**

Overall, the submission is visually polished and organized to effectively discuss their results and methods. However, I noticed several typos and grammatical errors throughout. Below are a few examples:

•	Table 3: `have or do not have` -> `have or have not`

•	Figure 6: Missing a panel `e`.

•	Line 131: `human annotators are shown with an action scene about` -> `human annotators are shown an action scene`

•	Line 132: `they can intuitively reasoning like` ->

•	Lines 194-195: `dimensions most differently or consistently` -> ?

•	Lines 300-301: `proven effective in action recognition tasks benefit by its` -> ?

•	Line 103, 170, 339, 845: The use of `Besides` as a sentence transition does not quite make sense.


Considering writing style, some long sentences may benefit if reworded or broken up. Examples include:

•	Lines 31-34.

•	Lines 121-124.

•	Lines 127-128 – the rationale (a).

•	Lines 136-138.

•	Lines 299-302.

Please continue to have your text proofread.

**Correctness:**

The collection methodology and assessment of data quality appear to be thorough and sound. However, the paper does not have any p-values or confidence intervals to support their comparisons of methods and benchmarks (Figures 3, 5, and 6. And Table 5).

**Documentation:**

Data documentation appears sufficient.

**Ethics:**

The authors sufficiently discuss ethical data collection in the Supplement. However, the ethical discussion of the research itself appears to largely discuss the benefits to AIGV research and not necessarily to society as a whole. There are several negative (fake impersonations, identity theft, etc.) as well as positive (the need to consider human perception and understanding to better understand generative models and enable more predictable behavior) with this work that may be more appropriate for this section.

**Limitations:**

A limitation that appears to be missed is the effort and labor required to obtain this dataset. This precludes the evaluation of new AIVG models directly with those presented in this paper. In that sense, it appears that while this dataset can benchmark both current and future action quality metrics, it can only be used to evaluate the models presented in this paper. Could this be clarified in the text?

**Opportunities For Improvement:**

•	Although the syllogism of actions is a clever decomposition of action quality and is intuitively reasonable, the motivation of this particular decomposition is somewhat lacking. I may be confused by some of the language in section 3.2 (see clarity below).

•	The three perspectives appear to be highly correlated throughout the results. As the paper currently stands, the benefit of this decomposition is unclear. There may be a missed opportunity to further explore/discuss the instances in which there are inconsistencies between perspectives.

•	Are there existing automated AQA, action-related metrics, VQA, or video-text alignment metrics that can be combined (e.g. linearly) to create more holistic methods that more closely recapitulate the GAIA perception results? There seems to be a non-trivial amount of correlation between GAIA and many of the metrics in Table 5. Is it possible that some of these metrics are correlated with distinct aspects of human perception? This may demonstrate a significant benefit of this dataset.

**Relation To Prior Work:**

To the best of my knowledge, this paper is not missing any major models or metrics in its discussion and comparison.

**Summary And Contributions:**

The authors are motivated by the challenge of assessing action quality in action-generative video models. They take the approach of collecting human-centered perceptions of action qualities in three categories from current video-generation models. Using their collected data, they evaluate current models and benchmarks and show the insufficiency of current benchmarks to holistically recapitulate human action understanding.

---

> ### Author Rebuttal · Authors · 2024-08-15
>
> We thank the reviewer for all the constructive suggestions, and we hope the following clarifications can address the reviewer's concerns:
>
> **P-values or confidence intervals.**  We added statistical tests as suggested to further support the obtained conclusions:
>
> - Fig.6: The MOS of complex actions (jumping/throwing) are lower than actions with small movements (e.g., communication, touching person, and using tools) (p<0.01  Two-side T-test)
> - Tab.5: We further conduct a T-test with a 95% confidence level to assess the statistical significance of the performance difference between any two methods, shown in Fig. 1 and Tab.1 of uploaded PDF file.
>
> **Motivation of action decomposition.** We merged this problem with the similar question raised by Reviewer T8oF. Please refer to the answer in the rebuttal for **Reviewer T8oF**: “**The necessity of analyzing ….” and “Clarify how ...”**
>
> **Discussion about the extreme instances.** We show two extreme instances in Fig. 1(c) in the main paper. The first one is an action about “surfing crowd” with severe subject artifacts but shows well on the action-scene interaction dimension (a reasonable splash of water). The second one is a “pumping fist” action, where we can observe a high-quality athlete with hardly any movement changes. This further proves the necessity of action process decomposition.
>
> **Performance of combining existing metrics.** We test several different combinations of existing metrics for comparison. As shown in Tab. 2 of the uploaded PDF file, in most cases, the performance of the combined one is within the best performance of a single one. Surprisingly, we found a performance gain when combining different variants of CLIPScore. We hypothesize that this is due to the spatial feature compensation provided by the different convolutional kernel sizes. Moreover, using a combination of multiple methods is less efficient in practical applications. In the future, we will explore the structure of different models and investigate the possibility of fusing them at the module level in an end-to-end way to better predict the action quality.
>
> **Clarification about the limitation of the effort and labor required to obtain this dataset.** We admit that collecting such a dataset requires much effort and labor. To minimize its usage restrictions, we have open-sourced the GAIA dataset. We also plan to update the dataset to include new models and develop new continual learning-based VQA methods in the future. We encourage the community to explore alternative datasets or construct new ones to ensure that the benchmarking of AI-generated actions remains a dynamic and inclusive process.
>
> **Typos.** Thank you for your detailed feedback regarding the writing in this paper. We have carefully reviewed the entire document and made revisions to improve the clarity and readability, shown as follows:
> - We changed the wording in the caption of Tab.3, Line 131, and the use of **“**Besides**”** in Line 103, 170, 339, 845.
> - Line 132→*intuitively reason like*
> - Lines194-195→*two dimensions are most different or consistent*
> - We adjusted the layout of the legends in Fig. 6, see Fig.2 in the uploaded PDF file.
> - Lines31-34→*Second, the content discrepancies in those AQA videos are often subtle, as the action subjects typically perform similar actions within a consistent environment. Examples include swimming and diving in a natatorium or gymnastics in a gym, lacking the consideration of scene diversity.*
> - Lines127-128→*The visibility of the action in videos is greatly affected by the rendering quality of the action subject, which is a crucial element of visual saliency information.*
> - Lines 136-138→*This reasoning-form evaluation* is inherently aligned to human perception and can help in understanding how different parts of action are perceived by humans. *In addition, such action decomposition strategy can be transformed into learning objectives to design more accurate objective quality assessment algorithms.*
> - Lines 299-302→*Moreover, BVQA and SimpleVQA leverage the SlowFast model as their motion feature extractor. This model has demonstrated effectiveness in various action recognition tasks due to its dual pathway design, which captures both spatial semantics and motion information parallelly. However, it encounters problems when applied to AIGVs, primarily because of the limited frames.*
>
> **Ethics discussion.** Our work holds the potential for significant social impact, both positive and negative. We anticipate that this work will raise consideration of human perception and understanding in AI-generated actions to better understand generative models and enable more predictable behavior. Currently, the human preference and perceptual sensitivity to the quality of action along the whole action process still remains as an open problem. This work provides significant guidance on how to optimize video generation models to produce videos with more pleasant actions. Meanwhile, we acknowledge that this study could raise some safety and ethical concerns. One challenging aspect of text-to-video models is the generation of NSFW content (such as violent and pornographic content), which may be offensive or inappropriate for some viewers and can potentially foster illegal transactions. Although some video generation platforms like MoonValley, Morph and Stable Video have built-in safety filters that detect prompts with NSFW content, they can still be circumvented through prompt engineering [1]. Additionally, AI video generation technology can be exploited by criminals for fake impersonations and identity theft. Our study also highlighted that some AI-generated videos can convincingly mimic individual’s facial expressions and actions, thereby posing a latent threat to public safety and eroding public trust in social media.
>
> [1] Chen Y, Zou J Y. Twigma: A dataset of ai-generated images with metadata from twitter[J]. Advances in Neural Information Processing Systems, 2024, 36.

---

> > ### Comment · Reviewer_5JDo · 2024-08-16
> > **Response to Author Rebuttal**
> >
> > I thank the authors for their consideration of my review. As my main concern about statistical tests has been addressed, I am updating my score.
> >
> > I appreciate the authors' engagement with the idea of combining metrics. It seems that overall the performance decreases when combining multiple metrics. Can the authors comment on why this happens? Is it just that inconsistencies and overall low performance across different metrics create confusion when combining?
> >
> > The extra motivation for the decomposition of actions clarified my intuition about the method. However, the syllogism seems to have not been required for a majority of the core results in the paper. I think some highlighting discussion similar to the response to reviewer jQF6, **Performance of CLIPScore in action completeness perspective**, could benefit the paper. As the examples in Fig 1C are hard to see, it may also be valuable to put extra examples in the Appendix.
> >
> > Overall, this is a high-quality and thorough study with valuable contributions.

---

> > > ### Author Rebuttal · Authors · 2024-08-17
> > >
> > > We are happy to address your questions and really appreciate your willingness to raise the score! We greatly appreciate your feedback on our approach to combining metrics and the observation regarding the changes in overall performance.
> > >
> > > We further captured part of the results in Tab. 2 of the uploaded PDF and added extra evaluation results for discussion. For a fair comparison, the following results are obtained under zero-shot settings (SRCC/PLCC):
> > > | Method | Subject | Completeness | Interaction |
> > > |-------|-------|-------|-------|
> > > |Human Action|0.2453/0.2369|0.2895/0.2812|0.2861/0.2743|
> > > |Action-Score|0.2023/0.1823|0.2867/0.2623|0.2689/0.2432|
> > > |Flow-Score|0.1471/0.1541|0.0816/0.1273|0.1041/0.1309|
> > > |VSFA|0.1934/0.1917|0.1379/0.1322|0.1602/0.1658|
> > > |-------|-------|-------|-------|
> > > |VSFA+Human Action|0.0836/0.0790|0.0059/0.0142|0.0135/0.0096|
> > > |VSFA+Action-Score|0.2599/0.2531|0.3149/0.3046|0.3054/0.2939|
> > > |VSFA+Flow-Score|0.1309/0.1506|0.0714/0.1253|0.0914/0.1283|
> > > |-------|-------|-------|-------|
> > > |CLIPScore-B/16|0.3360/0.3314|0.3841/0.3777|0.3753/0.3632|
> > > |CLIPScore-B/32|0.3398/0.3330|0.3944/0.3871|0.3875/0.3821|
> > > |CLIPScore-L/14|0.3211/0.3156|0.3657/0.3574|0.3585/0.3426|
> > > |VSFA+CLIPScore-B/16|0.3782/0.3733|0.4014/0.3990|0.3984/0.3906|
> > > |VSFA+CLIPScore-B/32|0.4162/0.4120|0.4377/0.4355|0.4364/0.4288|
> > > |VSFA+CLIPScore-L/14|0.3651/0.3582|0.3826/0.3793|0.3821/0.3709|
> > > |VSFA+CLIPScore-B/16+CLIPScore-B/32|0.4004/0.3938|0.4361/0.4303|0.4308/0.4192|
> > >
> > > We can observe that:
> > >
> > >  1) Combining VSFA with “Human Action” or “Flow-Score” did not yield performance improvements, rather, it resulted in a decrease in SRCC/PLCC scores. We attributed it to different scales of predicted scores, since Flow-Score is an optical flow-based metric.
> > >  2) Adding VSFA with three variants of CLIPScore shows better SRCC/PLCC on all three perspectives compared to their single forms. As mentioned in the main paper and the rebuttal for Reviewer jQF6, VQA methods perform better on subject quality than action completeness and action-scene interaction perspectives, which is opposed to CLIPScore. Therefore, combining these two kinds of metrics could effectively improve the subjective consistency of results. This observation provides intuition for the future development of better AQA methods.
> > >  3) CLIPScore and its variants outperform the other methods under zero-shot settings. This result suggests that considering both spatial and textual features to better associate visual features with scene descriptions is helpful in predicting action quality.
> > >
> > > Furthermore, considering the readability of the content in Fig. 1(c), we will put extra examples in the Appendix as suggested in the final version.

---

### Official Review · Reviewer_sWxM · 2024-07-25
**Paper 455 review**

**Rating:** 8
**Confidence:** 4
**Correctness:** Yes.
**Clarity:** Yes.

**Review:**

The quality of this work is high, as evidenced by its rigorous methodology and comprehensive analysis. The construction of the GAIA dataset involved a large-scale subjective evaluation, ensuring robustness and reliability. The paper also includes a detailed comparison with existing methods, highlighting the unique challenges posed by AI-generated videos. The authors clearly articulate the motivation behind the study, the methodology used, and the results obtained. Figures and tables are effectively used to illustrate key points, and the structure of the paper allows for easy navigation through its different sections. The introduction of GAIA, a dataset specifically tailored for AI-generated action quality assessment, is novel. Additionally, the use of a causal reasoning framework to evaluate action quality is an innovative approach that sets this work apart from previous studies.

**Strengths:**

1. GAIA is a well-curated, large-scale dataset that fills a critical gap in the evaluation of AI-generated videos.
2. The paper provides a thorough evaluation of existing AQA and VQA methods, highlighting their shortcomings in the context of AI-generated videos.
3. The use of causal reasoning for action quality assessment is a novel and insightful approach.
4. The analysis of different T2V models provides valuable insights into their strengths and weaknesses.
5. The dataset is supported by a robust human annotation process, ensuring the reliability of the evaluation.

**Additional Feedback:**

Please improve based on the feedback provided. Overall, I think this is a good paper that could be great!

**Documentation:**

Yes.

**Ethics:**

Good discussion in appendix.

**Limitations:**

Discussed above.

**Opportunities For Improvement:**

1. While the dataset is comprehensive, the scope of actions covered might still be limited compared to the vast range of possible AI-generated actions.
2. Additionally, I think a paragraph has to be added in the related works section discussing methods that use the idea of actors or occlusions and talk about mitigating biases. For example, [1-4] need to be discussed to show how the work builds on these ideas. These also build on synthetically creating data and a section should be added that discusses these.

[1] Choi, J., Gao, C., Messou, J.C. and Huang, J.B., 2019. Why can't i dance in the mall? learning to mitigate scene bias in action recognition. Advances in Neural Information Processing Systems, 32. [2] Gowda, S.N., Rohrbach, M., Keller, F. and Sevilla-Lara, L., 2022, October. Learn2augment: learning to composite videos for data augmentation in action recognition. In European conference on computer vision (pp. 242-259). Cham: Springer Nature Switzerland. [3] Gorpincenko, A. and Mackiewicz, M., 2022, November. Extending temporal data augmentation for video action recognition. In International Conference on Image and Vision Computing New Zealand (pp. 104-118). Cham: Springer Nature Switzerland. [4] Roberto de Souza, C., Gaidon, A., Cabon, Y. and Manuel Lopez, A., 2017. Procedural generation of videos to train deep action recognition networks. In Proceedings of the IEEE Conference on Computer Vision and Pattern Recognition (pp. 4757-4767).

**Relation To Prior Work:**

Could be better, as mentioned in the improvement part.

**Summary And Contributions:**

The paper introduces GAIA, a comprehensive dataset specifically designed for assessing AI-generated actions. GAIA includes 9,180 videos from 18 text-to-video (T2V) models, rated on three dimensions: subject quality, action completeness, and action-scene interaction, based on a causal reasoning framework.

The paper highlights the limitations of existing action quality assessment (AQA) methods, which are predominantly trained on real-world video datasets and do not effectively translate to AI-generated content due to the inherent differences in video characteristics. GAIA addresses this gap by providing a large-scale, human-annotated dataset that captures a wide range of AI-generated actions, enabling a more accurate and nuanced evaluation of T2V models.

Key contributions of the submission include the creation of a novel dataset (GAIA), the introduction of a new evaluation framework based on causal reasoning, and the benchmarking of existing AQA methods, action-related metrics, and video quality assessment (VQA) methods against human opinions. The findings reveal that traditional AQA and VQA methods correlate poorly with human evaluations, underscoring the need for more specialized assessment tools for AIGVs.

Overall, the study aims to catalyze progress in developing more effective AQA methods for AI-generated videos and provides substantial insights into the evaluation of action quality in these emerging technologies.

---

> ### Author Rebuttal · Authors · 2024-08-13
>
> We are grateful to the reviewer for acknowledging the significance of our findings and contributions! We hope the following clarifications can address the reviewer's concerns.
>
> **The scope of actions.** While the current dataset includes a wide range of AI-generated actions, we acknowledge that it represents only a subset of the possible actions AI could generate (limitations in Sec. 5). To balance the number of annotated videos (labor) with their availability and coverage as much as possible, we mainly focused on three specific aspects (i.e., whole-body actions, hand actions, and facial actions) during its construction. We plan to expand the dataset in the future to include a broader range of AI-generated actions. We also aim to involve the community in contributing to this effort, which will help in capturing a more comprehensive set of actions.
>
> **Extra discussions about the methods that use the idea of actors or occlusions.** Thanks for the novel view. We will refer and discuss these works in the context of action quality assessment in the related works. Here, we provide a consolidated view:
> - Human activities often occur in specific scene contexts, e.g., swimming in a swimming pool. However, it is also possible to enjoy champagne in the pool. Training an action quality assessment model for more diverse AI-generated actions using existing video datasets thus inevitably introduces such bias, which may opposed to paying attention to the actual action in the scene. Choi et al. [1] proposed to mitigate scene bias by adding an adversarial loss for scene types and masking out the human actors. Similar operations include extracting the foreground and background parts of the video as data augmented pairs to improve the accuracy of action recognition [2]. Apart from pixel space augmentation, Gorpincenko et al. [3] extended it further to utilize the time domain to perform deeper levels of temporal perturbations, thus improving the robustness of action classifiers. The above studies illustrate the necessity of disentangling action process from scenes or decomposing the action process to mitigate the effect of representation bias.
> - From the perspective of data provenance, virtual worlds and game engines are plausible techniques to generate editable actions as synthetic data before the Generative AI Era. Such synthetic data has been used to train visual models for lots of computer vision tasks (e.g., object detection, recognition, pose estimation, and scene understanding) to extract visual priors. Desouza et al. [4] proposed a diverse, realistic, and physically plausible dataset of human action videos using virtual world simulation software. Experiments show that mixing both synthetic and real samples at the mini-batch level during training can significantly improve action recognition accuracy. Similarly, the controllability of both the type and quality of AI-generated actions (using different types of prompts and video generation models) offers a feasible solution for constructing large-scale action quality assessment datasets, especially for those irrational scenarios.
>
> [1] Choi, J., Gao, C., Messou, J.C. and Huang, J.B., 2019. Why can't i dance in the mall? learning to mitigate scene bias in action recognition. Advances in Neural Information Processing Systems, 32.
>
> [2] Gowda, S.N., Rohrbach, M., Keller, F. and Sevilla-Lara, L., 2022, October. Learn2augment: learning to composite videos for data augmentation in action recognition. In European conference on computer vision (pp. 242-259). Cham: Springer Nature Switzerland.
>
> [3] Gorpincenko, A. and Mackiewicz, M., 2022, November. Extending temporal data augmentation for video action recognition. In International Conference on Image and Vision Computing New Zealand (pp. 104-118). Cham: Springer Nature Switzerland.
>
> [4] Roberto de Souza, C., Gaidon, A., Cabon, Y. and Manuel Lopez, A., 2017. Procedural generation of videos to train deep action recognition networks. In Proceedings of the IEEE Conference on Computer Vision and Pattern Recognition (pp. 4757-4767).

---

> > ### Comment · Reviewer_sWxM · 2024-08-20
> > **Response to rebuttal**
> >
> > Please do include a more comprehensive related works section as the topics are extremely correlated. Overall this is a great paper and I have improved my rating after taking into account the other reviews as well.

---

> > > ### Author Response · Authors · 2024-08-20
> > > **Thank the reviewer**
> > >
> > > We would like to thank the reviewer for all the valuable comments and questions. We are grateful for your engagement in the rebuttal process. We will further refine the related work in the final version.

---

### Author Response · Authors · 2024-08-15
**Looking forward to further discussions to address concerns**

We would like to thank all reviewers for their valuable comments. We hope our responses have adequately addressed your previous concerns. We take this as a great opportunity to improve our work and shall be grateful for any additional feedback you could give us.

---

### Decision · Program_Chairs · 2024-09-26

**Decision:**

Accept (Spotlight)

**Comment:**

The paper introduces GAIA, a dataset specifically designed for assessing AI-generated actions, addressing the limitations of existing action quality assessment methods. The dataset includes videos from 18 text-to-video models rated on three dimensions based on a causal reasoning framework. The study benchmarks current AQA methods against human evaluations and highlights the need for specialized assessment tools for AI-generated videos. The authors aim to advance AQA methods for AI-generated content and provide insights into evaluating action quality in emerging technologies. Overall, the paper introduces a novel use of AQA in assessing generated videos and presents a valuable contribution to the field.